



# Impact of water mass dynamical and property variability on the inflow of a semi-enclosed sea.

Becca Beutel[1], Susan E. Allen[1], Jilian Xiong[2], and Maite Maldonado[1]

[1]Department of Earth, Ocean and Atmospheric Sciences, University of British Columbia, Vancouver, BC, Canada
[2]School of Oceanography, University of Washington, Seattle, WA, USA

**Correspondence:** Becca Beutel (rbeutel@student.ubc.ca)

**Abstract.** The biogeochemistry of the Salish Sea is strongly connected to its Pacific Ocean inflow through Juan de Fuca Strait (JdF), which varies seasonally and interannually in both volume and property flux. Long-term trends in warming, acidification, and deoxygenation are a concern in the region, and inflow variability influences the flux of tracers potentially contributing to these threats in the Salish Sea. Using ten years (2014-2023, inclusive) of Lagrangian particle tracking from JdF, we quantified

the contributions of distinct Pacific water masses to interannual variability in JdF inflow and its biogeochemical properties. We decompose variability in salinity, temperature, dissolved oxygen, nitrate, and carbonate system tracers into components arising from changes in water mass transport (dynamical variability) and changes in source properties (property variability). Observations in the region provide insight into water mass processes not resolved by the model, including denitrification and trace metal supply. Deep water masses dominate total inflow volume and drive variability in nitrate flux through changes in

transport. Shallow water masses, particularly south shelf water, exhibit greater interannual variability and disproportionately affect temperature, oxygen, and [TA–DIC], driving change through both dynamical and property variability. This study high-lights the combined roles of circulation and source water properties in shaping biogeochemical variability in a semi-enclosed sea, and how these roles differ between biogeochemical tracers. It provides a framework for attributing flux changes to specific source waters and physical and biogeochemical drivers, with implications for forecasting coastal ocean change under future

climate scenarios.

## 1 Introduction

Coastal regions are critical interfaces between land and ocean, supporting diverse ecosystems and large population density (Cosby et al., 2024). Despite their accessibility compared to the open ocean, coastal oceanography remains challenging due to the high spatial and temporal variability of these areas. The resolution possible in world ocean models is too low to capture

the high spatial variability in coastal areas, including intricate bathymetry, riverine inputs (e.g., Thomson et al., 1989), or the coastal-trapped waves that can impact nearshore dynamics and properties far afield (Brink, 1991; Engida et al., 2016). The physical, chemical, and biological conditions in coastal areas are inextricably connected to large scale processes, yet exhibit spatial and temporal variability orders of magnitude greater than the open ocean (e.g., Chang and Dickey, 2001; Fassbender



et al., 2018). Evaluating the extent to which offshore processes influence coastal regions is necessary for estimating the impacts of projected changes to coastal circulation and properties.

Water masses, distinct oceanic bodies defined by their properties, serve as tools for tracing connectivity and biogeochemical transport across regions. There is evidence of Pacific Equatorial Water (PEW), for example, 11,000 km north of its equatorial origin (Thomson and Krassovski, 2010). While often defined according to their salinity and temperature, water masses are biogeochemical conduits; in addition to being warm and high in salinity, the PEW is an important nutrient source and has been linked to hypoxic conditions far afield (Thomson and Krassovski, 2010; Bograd et al., 2008). Variations in water mass dynamics (e.g., Huyer et al., 2007; Jutras et al., 2020) and their properties (e.g., Jacox et al., 2024; Kurapov, 2023) —both interannual and long-term— can significantly impact the biogeochemical conditions of coastal areas.

The Salish Sea (Fig. 1), a semi-enclosed sea in the Northeast Pacific Ocean (Sec. 2), provides an ideal case study for examining the relationship between water mass variability and coastal biogeochemistry. This densely populated region is strongly influenced by oceanic inflows that vary seasonally in origin and properties (Beutel and Allen, 2024; Brasseale and MacCready, 2025; Masson, 2006). Interannual variability in shelf and Salish Sea inflow properties (Stone et al., 2018; Alin et al., 2024; Beutel and Allen, 2024) and in the region's ecological composition (Li et al., 1999; Del Bel Belluz et al., 2021), point to deviations from the typical seasonal cycle, changes in source water characteristics, or both. Understanding the mechanisms that drive this variability is particularly urgent as long-term climate trends alter both circulation patterns and the biogeochemical properties of Pacific source waters.

This study investigates how biogeochemical variability in the Salish Sea connects to the dynamical and property variability of inflowing water masses over a period of ten years. Through a combination of model output and observations, we attempt to attribute biogeochemical changes to specific water masses and their modes of interannual variability. By doing so we provide insights into the mechanisms driving variability in this coastal system and discuss the implications under a changing climate.

## 2 Regional Overview

The Salish Sea's largest connection to the Pacific, Juan de Fuca Strait (JdF), functions like a fjord-like estuary with dense Pacific inflow below brackish strait outflow. Exchange is driven by the discharge of the region's many rivers and the salinity difference between the Sea and the adjacent shelf waters and is sensitive to tides, winds, and gravitational circulation (MacCready and Geyer, 2024). Inflow through JdF is the largest source of many biologically significant constituents to the Salish Sea, such as nutrients like nitrate ($NO_3$; Mackas and Harrison (1997); Khangaonkar et al. (2012),) dissolved inorganic carbon (DIC) and total alkalinity (TA;Jarnikova et al. (2022)), trace metals like cadmium (Cd; Kuang et al. (2022)), and temperature anomalies (Khangaonkar et al., 2021). These constituent loads exhibit significant temporal variability. For instance, nitrate inflow concentrations, observed over ten years, had a standard deviation equal to 65% of the mean annual load (Sutton et al., 2013). Variations in JdF inflow loads are influenced by a complex interplay of local and remote shelf dynamics, as well as large-scale Pacific circulation.





Typical seasonal variation on the shelf near the entrance to JdF is directly linked to wind forcing (Brasseale and MacCready, 2025; Beutel and Allen, 2024). Poleward wind drives downwelling in the region, and local water primarily originates from the southern shelf and continental slope. Under strong poleward winds, outflow from the Columbia River, located 200 km to the south of the mouth of JdF, can also contribute to the inflow (Giddings and MacCready, 2017). Winds shift to upwelling

favourable in the summer months (starting April 9 $\pm$29 days and ending October 17 $\pm$17 days; Hourston and Thomson (2024)). During upwelling, water originates from the northern shelf and offshore, generally within the top 300 m of the water column (Beutel and Allen, 2024). The contrasting properties of downwelled (fresher, more oxygen-rich, and nutrient- and DIC-poor) and upwelled water account for much of the seasonal variability in JdF inflow (Masson, 2006). However, these differences alone do not explain interannual variability.

The Salish Sea is located at the northernmost end of the California Current System (CCS), an eastern boundary current system located between the North Pacific Gyre and the western coast of North America, spanning $\sim 50°$N (Northern Vancouver Island, Canada) to $\sim 15 - 25°$N (Baja California, Mexico). However, the latitude of the northern limit of the CCS fluctuates due to variations in the location and strength of the Aleutian Low, which pushes the CCS southward during winter (Thomson, 1981) and during positive phases of the Pacific Decadal Oscillation (PDO; Zhang and Delworth (2016)), adding to interannual

variability in the northern CCS. The currents in this system include the California Current (CC), the California Undercurrent (CUC), the Shelf-Break Current, the Davidson Current, and the Columbia River Plume. The strength, depth, and spatial extent of these currents have large implications for the productivity and health of the Northeast Pacific coast. Variability in the CCS, driven by remote wind or current strength and manifesting in a change in temperature and nutrient conditions, have been shown to influence the distribution and abundance of phytoplankton, the organisms that depend on them, and the reproductive success

of many species (Checkley and Barth, 2009).

The CC is the eastern arm of the North Pacific Gyre, fed by the North Pacific Current (Hickey, 1998). The CC is a broad and shallow current, flowing equatorward year-round in the top 250 m and within 1000 km of the coast (Auad et al., 2011; Checkley and Barth, 2009). Interannual variability in the strength of the North Pacific Current and its relative contributions to the CC versus the Alaska Current can alter the CC's properties and strength (Checkley and Barth, 2009; Cummins and Freeland,

2007). The CC predominantly carries Pacific Subarctic Upper Water (PSUW), a relatively cold ($3° - 15°$ C), fresh (32.6–33.7 psu), and oxygen-rich water mass originating from the surface waters of the North Pacific (Thomson and Krassovski, 2010; Bograd et al., 2019). Its core is situated at the 25.8 $\sigma_t$ density surface (Bograd et al., 2019), approximately 100 m deep in the vicinity of Vancouver Island. During upwelling, wind driven equatorward flow on the continental shelf—the Shelf-Break Current—merges with the CC, which acts as the offshore extension of the Shelf-Break Current (Hickey, 1998; Thomson and

Krassovski, 2010). However, the Shelf-Break Current is fresher than the CC (Sahu et al., 2022) due to the large influence of coastal rivers (Stone et al., 2020).

The CUC is the opposing subsurface flow associated with eastern boundary regions, flowing poleward year-round along the continental slope at depths between 100 and 300 m (Thomson and Krassovski, 2010, 2015; Pierce et al., 2000). The CUC is relatively narrow, transporting a fraction of the transport of the broader CC (Thomson and Krassovski, 2015). Variations in

CUC transport, depth, and physical properties have been linked to remote wind forcing and coastal-trapped waves originating





from California and Oregon (Thomson and Krassovski, 2015; Engida et al., 2016). The CUC carries PEW, a relatively warm ($7° − 23°$ C), saline (34.5–36.0 psu), and nutrient-rich water mass originating from mixing in the equatorial Pacific (Thomson and Krassovski, 2010). The core of the CUC typically lies at the 26.55 $\sigma_t$ density surface (Bograd et al., 2019), just shallower than 200 m near Vancouver Island (Thomson and Krassovski, 2015). As it travels northward, the CUC undergoes mixing and sheds PEW, resulting in a composition of approximately 30% PEW and 70% PSUW by the time it reaches Vancouver Island (Thomson and Krassovski, 2010).

During downwelling, the Shelf-Break Current is replaced by the poleward flowing Davidson current, which transports southern shelf water to the region and spans farther offshore than the continental slope (Thomson and Krassovski, 2010; Giddings and MacCready, 2017). The Davidson current flows faster (Mazzini et al., 2014; Thomson and Krassovski, 2015) and at significantly shallower depths (surface to ∼200 m, the depth of the shelf-break), making it spatially distinct from the CUC as well as distinct in the properties it carries. Water transported by the Davidson current is colder and fresher than CUC water (Sahu et al., 2022), likely due to the influence of coastal river discharge (Mazzini et al., 2014).

## 3 Methods

This study utilizes a coupled physical-biogeochemical ocean model to integrate a widely used physical oceanographic technique— simulation-based Lagrangian flow (Sec. 3.2)—with observations of chemical tracers. This section outlines the model, describes the simulation and analysis techniques, and details the sources of the observational data.

### 3.1 Physical-Biogeochemical Ocean Model: LiveOcean

LiveOcean (Fig. 1) is a 3D physical-biogeochemical ocean model developed by the University of Washington Coastal Modelling Group (MacCready et al., 2021). The model uses the Regional Ocean Modelling System (ROMS) version 4.2 architecture (Haidvogel et al., 2000; Shchepetkin and McWilliams, 2005). A detailed description of the previous iteration of LiveOcean is provided in (MacCready et al., 2021) with updates to the model outlined in Xiong et al. (2024). Notable improvements to the new iteration of LiveOcean are separating dissolved organic nitrogen (DIN) into $NO_3$ and $NH_4$ (greatly improving $NO_3$ estimates), including precipitation and evaporation (decreasing the surface salinity error), making the vertical and horizontal advection scheme of biological tracers consistent with that of salinity and temperature, and accounting for more small rivers and including biogeochemical constituents in their outflow (Xiong et al., 2024). The version of LiveOcean used in this study was initialized on October $7^{th}$ 2012 and continues to run with the settings described below as of the submission of this article.

The model grid follows lines of constant longitude and latitude, with a horizontal resolution of approximately 500 m in most of the Salish Sea and along the Washington coast. Resolution gradually increases to 1500 m in the northern Strait of Georgia and 3000 m near the open boundaries, the lowest resolution in the boundaries of analysis in this study is 1650 m (Fig. 1). Vertically, the model is divided into 30 sigma layers, with a higher density of layers near the surface and bottom.

The conditions of all three open boundaries are based upon fields from the global HYCOM model (Metzger et al., 2014), with daily velocities, temperature, salinity, and sea surface height (ssh) smoothed to remove inertial oscillations. Biogeochemical





variables at the open boundaries are specified, using regressions against salinity, derived from cruise measurements (Feely et al., 2016). Tides along these boundaries are forced using eight tidal components (K1, O1, P1, Q1, M2, S2, N2, and K2)
from TPXO 9 (Egbert and Erofeeva, 2002). The model's bathymetry incorporates products described in Finlayson (2010) for Puget Sound, Sutherland et al. (2011) for the remainder of the Salish Sea, and Tozer et al. (2019) for the remainder of the model area, all smoothed for stability (Giddings et al., 2014). Daily average gauged flow from the United States Geographical Survey (USGS) and Environment Canada are used to force rivers. For ungauged watersheds, flow estimates are derived from nearby gauged rivers using scaling factors from Mohamedali et al. (2011). Tracer flux from rivers is specified based on local
climatology (Xiong et al., 2024). Atmospheric forcings are from output of a Weather Research and Forecasting (WRF) model run by Dr. Cliff Mass at UW (Mass et al., 2003), with a resolution of 1.4 km within most of the LiveOcean domain, 4.2 km north of 49° N or west of 126° W, and 12.5 km north of 50° N.

The updated iteration of LiveOcean performed well in evaluations along the shelf and slope from 2014 to 2018 as detailed in Xiong et al. (2024) and summarized here. In Supplement S1 we conduct separate evaluations for the subregions of the
model, as defined in Table 1. Modelled water properties were compared to data from the Washington Department of Ecology, Department of Fisheries and Oceans (DFO), and National Centers for Environmental Information (NCEI). Water below 20 m had a RMSE (bias) of 0.3 (-0.1) $\mathrm{g\,kg^{-1}}$ for salinity, 0.7 (0.0) $^{\circ}\mathrm{C}$ for temperature, 1.1 (-0.2) $\mathrm{mg\,L^{-1}}$ for dissolved oxygen (DO), 5.0 (0.0) $\mathrm{mmol\,m^{-3}}$ for $NO_3$, 1.6 (0.8) $\mathrm{mmol\,m^{-3}}$ for $NH_4$, 47.4 (13.5) $\mathrm{mmol\,m^{-3}}$ for DIC, and 25.5 (-2.7) for TA (Xiong et al., 2024). The high bias and RMSE of $NH_4$ relative to its range in the region (0-8 $\mathrm{mmol\,m^{-3}}$) led to its removal from
model analysis in this paper. Subregion evaluations (Supplement S1) revealed some differences in the bias among regions; notably, deep water masses exhibit a smaller bias in all properties compared to the shallow water masses, and northern waters overestimate DO while all other regions underestimate it.

### 3.2   Lagrangian Tracking with Ariane Model

Lagrangian ocean analysis tracks the movement of free-moving entities to estimate ocean pathways by applying the Lagrangian
lens of fluid dynamics (Bennett, 2006). Using time-varying velocity, and tracer fields from an ocean model, virtual particles can be simulated to behave like objects (e.g., Van Sebille et al., 2018), zooplankton (e.g., Brasseale et al., 2019), environmental DNA (e.g., Xiong et al., 2025), and pollutants (e.g., Sayol et al., 2014). In this study, the particles are simply neutrally buoyant parcels of water. Their paths and transports serve as proxies for dynamics. Particles can be tracked backwards in time, allowing for source water analysis while avoiding biases inherent in seeding particles from an expected source direction.
Ariane is an offline Lagrangian tracking algorithm that assumes time-varying velocity changes linearly between opposite cell faces, preserving local three-dimensional non-divergence in the flow and enabling backward tracking (Blanke and Raynaud, 1997; Van Sebille et al., 2018). This assumption means subgrid-scale mixing is not directly included within Ariane but is parameterized within the underlying numerical model (LiveOcean). Given LiveOcean's high resolution within the study domain, large turbulent eddies are resolved (Stevens and Pawlowicz, 2023), so the lack of explicit subgrid-scale mixing does
not significantly impact transport results (Allen et al., 2025). Ariane supports two analysis modes: qualitative, which tracks





individual parcel trajectories, and quantitative, which seeds more particles to enable volume transport analysis without saving individual trajectories. This study employed backward tracking in the quantitative mode.

In the quantitative mode, parcels are seeded along an "initialization" section (brown boundary in Fig. 1) and tracked until they reach simulation boundaries (pink boundaries in Fig. 1) or pass back over the initialization section (hereafter referred to as loop(ed) parcels) (Blanke et al., 2001). The initialization and simulation boundaries are distinct from model boundaries, they are user-defined and must follow the model grid (i.e. they cannot cut diagonally across cells). Parcels that never reach a boundary within the analysis period are classified as "lost." Parcels are distributed across the initialization section proportional to the transport (where transport through a cell $q$ is equal to the velocity through a cell multiplied by its area, $q = u \times A$) in each model grid cell at each time-step. The number of parcels seeded, $n$ per cell, is determined by dividing $q$ by the user-defined maximum transport per parcel $q_{max}$ ($n = q/q_{max}$, rounded up to the integer). Parcels are evenly distributed within the cell, and their constant volume flux $V_n$ ($\mathrm{m^3\,s^{-1}}$) is defined as $V_n = q/n$.

Simulations were conducted from 2014 to 2023 (inclusive), with separate runs for each upwelling, downwelling, spring transition, and fall transition period (as defined in Sec. 3.4). Particles were continuously released over the analysis period, with each run including an additional 100 days without particle seeding to allow particles sufficient time to travel between boundaries (Beutel and Allen, 2024). Six tracers from LiveOcean were input into the simulations: salinity, temperature, DO, nitrate, DIC, and TA; necessitating three runs per analysis period as only two tracers can be input into Ariane at a time. It should be noted that [TA-DIC] was used to study carbonate chemistry in JdF inflow, as opposed to TA and DIC individually, as the difference between the two terms can be used more effectively to investigate ocean acidification (OA; Xue and Cai (2020)); a large positive [TA-DIC] indicates a high buffering capacity with respect to OA, while low or negative values indicate that this region is vulnerable to OA.

Looped parcels that pass back over the initialization section within 24 hours of seeding are considered "tidally pumped" parcels and are removed from analysis, the remaining looped parcels can be thought of as reflux circulation through JdF (MacCready et al., 2021). Beyond lost and looped parcels, sources were assigned based on parcel position and salinity at the point they crossed a boundary (Table 1). The analysis boundaries follow those defined in Beutel and Allen (2024): the initialization section, south, north, and offshore, with salinity and depth thresholds redefined for LiveOcean output (Supplement S2). The location of the initialization section is set inland of the mouth of JdF to reflect waters that actually reach the sea's inner basins, or at least do to the degree found in Beutel and Allen (2024). While the majority of south brackish water is expected to originate from the Columbia River plume (Hickey et al., 2009), evaluating the dominance of that source on brackish flow from the south is outside of the domain and scope of this study; other small rivers along the coast may contribute to this source. Offshore water is divided into surface and deep water masses based on the approximate offshore mixed-layer depth (Supplement S2.2).



**Table 1.** Source water definitions for the particle tracking simulation and the division of observations. The boundaries for particle tracking definitions refer to those shown in Fig. 1. The regions (offshore, slope, shelf) of observations are defined by their location relative to bathymetric contours, offshore is oceanward of 2000 m, shelf is shoreward of 200 m, and slope is between the two. Note that no observations exceeding depths of 500 m were used.

| Source | | Definition |
|---|---|---|
| **Particle Tracking Simulation** | | |
| Shallow waters | North Shelf | North boundary |
| | Offshore Surface | Offshore boundary, depth $\leq 120\,\mathrm{m}$ |
| | South Brackish | South boundary, salinity $< 32\,\mathrm{g\,kg^{-1}}$ |
| | South Shelf | South boundary, $32\,\mathrm{g\,kg^{-1}} <$ salinity $\leq 33.5\,\mathrm{g\,kg^{-1}}$ |
| Deep waters | CUC | South boundary, salinity $\geq 33.5\,\mathrm{g\,kg^{-1}}$ |
| | Offshore Deep | Offshore boundary, depth $> 120\,\mathrm{m}$ |
| | Loop water | Initialization boundary, transit time $> 24$ hours |
| **Observations** - depth $< 500\,\mathrm{m}$ | | |
| Shallow waters | North Shelf | Shelf and slope water, latitude $\geq 49°\,\mathrm{N}$, salinity $> 31.5\,\mathrm{g\,kg^{-1}}$, depth $\leq 200\,\mathrm{m}$* |
| | Offshore Surface | Offshore water, depth $< 120\,\mathrm{m}$ |
| | South Brackish | Latitude $< 47.3\,°N$, salinity $\leq 31.5\,\mathrm{g\,kg^{-1}}$ |
| | South Shelf | Shelf and slope water, latitude $\leq 47.3°\,\mathrm{N}$, $31.5 <$ salinity $< 33.7\,\mathrm{g\,kg^{-1}}$, depth $\leq 200\mathrm{m}$ |
| Deep waters | CUC | Slope water, latitude $\leq 50.5°\,\mathrm{N}$, salinity $\geq 33.7\,\mathrm{g\,kg^{-1}}$* |
| | Offshore Deep | Offshore water, depth $\geq 120\,\mathrm{m}$ |
| | Domain | Shelf and slope water, $47.3 <$ latitude $< 49\,°N$ |

*All observations north of $49°\,\mathrm{N}$ and shallower than 200 m were less saline than $33.7\,\mathrm{g\,kg^{-1}}$, as such the CUC and North water observation definitions do not overlap.

## 3.3 Observed Tracers

Observations on and offshore of the BC, Alaska, Washington, and Oregon coasts were collated from nine sources (Table 2). These datasets span a wide range of time (1930 to the end of 2023), spatial coverage, and measurement techniques, including cruise-based bottle and/or conductivity temperature depth (CTD) profiles as well as CTD-mounted moorings.

Datasets were standardized to consistent units: absolute salinity ($S_A$) in $\mathrm{g\,kg^{-1}}$, in-situ temperature in $°C$, DO in $\mu\mathrm{mol\,kg^{-1}}$, density and spiciness in $\mathrm{kg\,m^{-3}}$, nutrients ($NO_3$, dissolved inorganic phosphate (DIP), dissolved silicon (DSi)), and tracers indicating or involved in nitrification/denitrification ($N^*$, nitrite ($NO_2$), ammonium ($NH_4$)) in $\mathrm{mmol\,m^{-3}}$, carbon chemistry parameters (TA, DIC, [TA-DIC], carbonate ($CO_3^{2-}$)) in $\mu\mathrm{mol\,kg^{-1}}$, and trace metals (Cd, Cobalt (Co), Copper (Cu), Iron (Fe), Manganese (Mn), Nickel (Ni), Zinc (Zn)) in $\mathrm{nmol\,kg^{-1}}$. Note that aragonite ($\Omega_{ar}$) and calcite saturation ($\Omega_{cal}$) are unit-less. Observations of density, spiciness (McDougall and Krzysik, 2015; Pawlowicz et al., 2012), and $N^*$ (Gruber and Sarmiento, 1997) were calculated where the necessary parameters were available. Outliers were removed by excluding data points that lay more than four standard deviations from the variable's mean. An exception was made for salinity, where values far below the mean but greater than zero were retained to preserve freshwater measurements. To reduce the computational load for large



datasets with unnecessarily high resolutions (e.g., moorings with minute-by-minute measurements), datasets were binned into day-averages at 1 m depth intervals.

After combining datasets, duplicates were identified and removed based on latitude (to two decimal places), longitude (to two decimal places), time (to the day), and depth (1 m bins). Duplicate observations were combined by taking the mean of the measurements. To encompass the northern CCS, only observations shallower than 500 m, east of 145.5 °W, and between 40

and 50.8 °N were kept (Checkley and Barth, 2009). While deep water masses may influence the Salish Sea via upwelling onto the shelf or through the Juan de Fuca Canyon, upwelled water originates from depths shallower than 500 m with the bulk from depths less than 150 m (Beutel and Allen, 2024; Bograd et al., 2001).

Observations were categorized as offshore, slope, or shelf based on the bathymetry of the observation sites (Fig. 1). Offshore observations were defined as those seaward of the 2000 m isobath, shelf as those landward of the 200 m isobath, and slope as

those between these two boundaries. Further division into sources (Table 1) was based on trajectories identified in this study (Sec. 4.1) and Beutel and Allen (2024), and property-property diagrams of temperature, $S_A$, $NO_3$, and TA. The sensitivity of the source classifications to these criteria is discussed in Supplement S2.





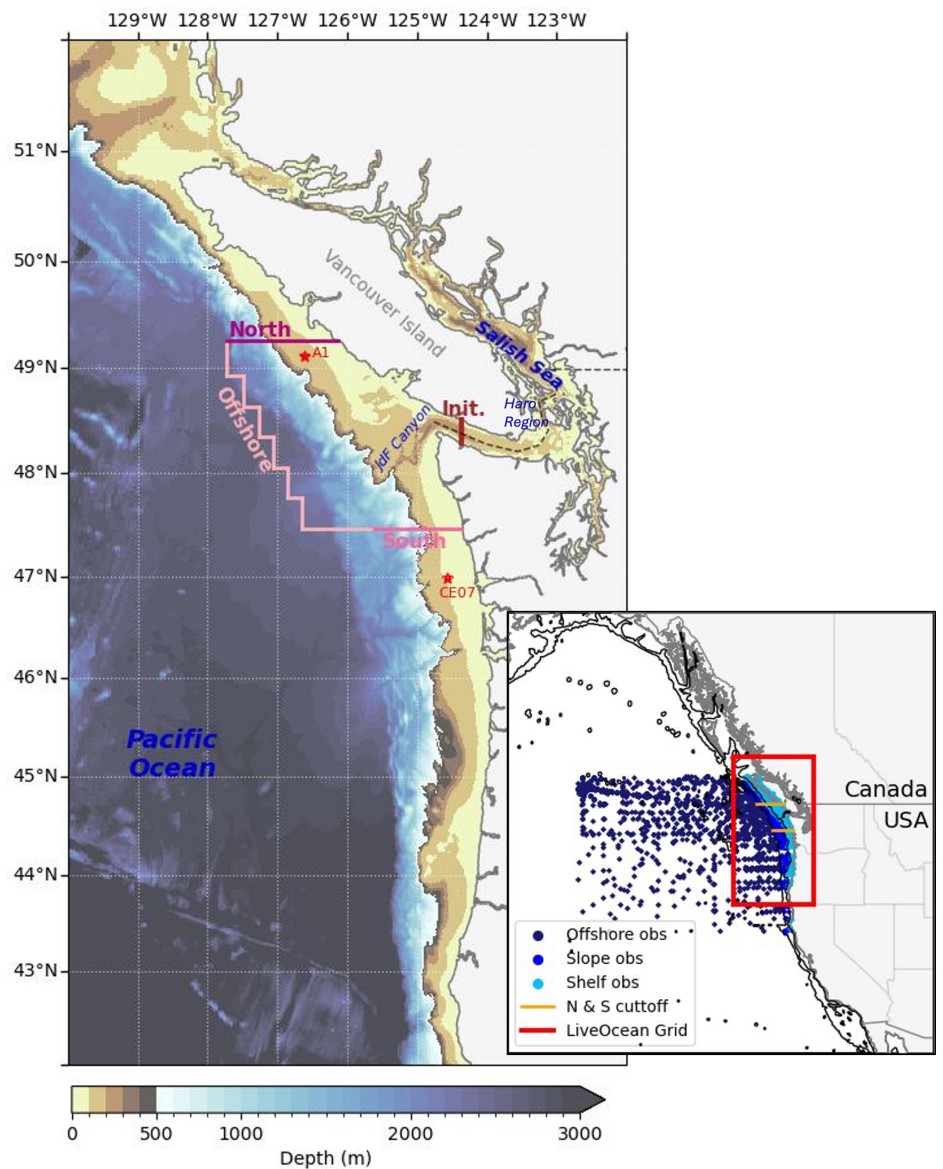

**Figure 1.** Map of the the LiveOcean domain with bathymetry. The Salish Sea is located along the coast of British Columbia and Washington (the division between Canada and United States, $\sim 49°$N, is shown by a dashed black line in the main map), and is enclosed by Vancouver Island. The simulation initialization boundary, inland of the mouth of JdF, is shown by a brown line, while the domain boundaries (North, Offshore, and South) are shown in pink. The location of moorings employed to determine the upwelling and downwelling run-dates (A1 and CE07) are highlighted by red stars. The inset map shows the boundaries of LiveOcean (red box) in the larger context of the Northeast Pacific Ocean. Observations used for water mass definitions are divided based on 200 m and 2000 m isobars, in black, with the offshore observations in navy, slope in blue, and shelf in cyan. The cutoff between the north and south areas of analysis are shown with orange lines over the shelf and slope (Table 1).



**Table 2.** Summary of collated observations (Fig. 1 1). The description of the dataset and number of observations reflects what was used in this study post-processing and is not a full reflection of what is available within that dataset. The inclusion of a variable in a dataset does not necessarily mean that it was measured at every site – the number of observations per variable, divided into the regions defined in Table 1, is provided in Table A1.

| Dataset Title | Variables | Observations (unique) | Description | Source |
|---|---|---|---|---|
| GEOTRACES | temperature, salinity, DO, $NO_3$, $NO_2$, $NH_4$, DIP, DSi, Cd, Co, Cu, Fe, Mn, Ni, Zn | 542 (383) | Bottle samples of trace elements along Line P and Gulf of Alaska cruises from 2012-2020. | GEOTRACES Intermediate Data Product Group (2023); Taves et al. (2022) |
| Institute of Ocean Sciences (IOS) Moored CTD Data | temperature, salinity, DO | 185,233 (185,224) | CTDs mounted on moorings along the BC coast and shelf between 2008 and 2023. | (Department of Fisheries and Oceans Canada, 2024c) |
| IOS CTD Profile Data | temperature, salinity, DO | 63,415 (62,649) | Profiles taken with CTDs mounted on Rosettes alongshore and offshore (cutoff at 145°W) of the BC, Alaska, Washington, and Oregon coast between 1965 and 2023. | (Department of Fisheries and Oceans Canada, 2024b) |
| IOS Rosette Bottle Data | temperature, salinity, DO, $NO_3$, DIP, Dsi | 154,091 (152,751) | Niskin bottle samples alongshore and offshore (cutoff at 145°W) of the BC, Alaska, and northern Washington coast between 1930 and 2023. Data resampled in time (days) and depth (10 m bins). | (Department of Fisheries and Oceans Canada, 2024a) |
| National Centre for Environmental Informatics (NCEI) Historical Pacific Northwest (PNW) Data | temperature, salinity, DO, $NO_3$, DIP, DIC, TA | 1,386 (977) | Discrete bottle observations from the R/Vs Endeavour, CCGS John P. Tully and Parizeau Line P cruises from 1985 to 2017 | Miller et al. (2013) |
| NCEI Coastal Ocean Data Analysis Product in North America | temperature, salinity, DO, $NO_3$, $NO_2$, $NH_4$, DIP, DSi, DIC, TA, $CO_3^{2-}$, $\Omega_{ar}$, $\Omega_{cal}$ | 6,138 (6,128) | Data from the West Coast Ocean Acidification (WCOA) cruises and other west coast of North American cruises with carbon data from 2011-2017 and 2021. | Jiang et al. (2021) |
| Newport Hydrographic Line (NHL) | temperature, salinity, DO | 500,083 (500,073) | CTD data collected bi-weekly at the NHL off Oregon from 1997-2021. | Risien et al. (2022) |
| Ocean Observatories Initiative (OOI) Coastal Endurance: Washington | temperature, salinity, DO, $NO_3$ | (50,567) | Two lines of three near surface moorings at 44.6 and 47.0°N, with hourly measurements at the coast, in the mid-shelf, and over the continental slope from 2014 to 2023. | Ocean Observatories Initiative (2024) |
| Olympic Coast National Marine Sanctuary (OCNMS) Cruise data | temperature, salinity, DO | (46,231) | CTD data collected within the OCNMS region (northern end of the Pacific Coast of Washington) from 2004-2023. | Risien et al. (2024) |



### 3.4 Upwelling and Downwelling Timing

Upwelling and downwelling were divided in some of the analysis in this study to identify the impact of seasonality, but most
analysis was conducted over a full year. Please note that when the text refers to annual values it does not mean one calendar year,
instead a year in this paper refers to the combination of one set of consecutive downwelling, spring transition, upwelling, and
fall transition periods, all of which differ in length interannually. A combination of upwelling estimates were used to identify
the length of these periods: alongshore velocity measurements at moorings A1 and CE07 (Fig. 1a, maintained by the DFO
and the Ocean Observatories Initiative (OOI), and available from 2013-2020 and 2015-present, respectively), spring and fall
transition timing (upwelling showed with yellow bars in Fig. 2b, Hourston and Thomson (2024), available from 1980-present),
and the Bakun Index at $48°$ N (upwelling showed with grey bars in Fig. 2, Bakun (1973); Bograd et al. (2009), available from
1967-present). Upwelling and downwelling timing (red and blue in Fig. 2a, respectively) was determined by the overlap of
these measures, with non-overlapping periods designated as transition intervals (white in Fig. 2a). If this method produced
buffer periods shorter than 20 days (Lynn et al., 2003), or if only one upwelling estimate was available (observations predating
estimates in Hourston and Thomson (2024)), the bounding upwelling and downwelling periods were shortened to ensure a
minimum transition length of 20 days. For periods predating the Bakun Index (before 1967), upwelling and downwelling
periods were conservatively set to May–August (inclusive) and November–February (inclusive), respectively.

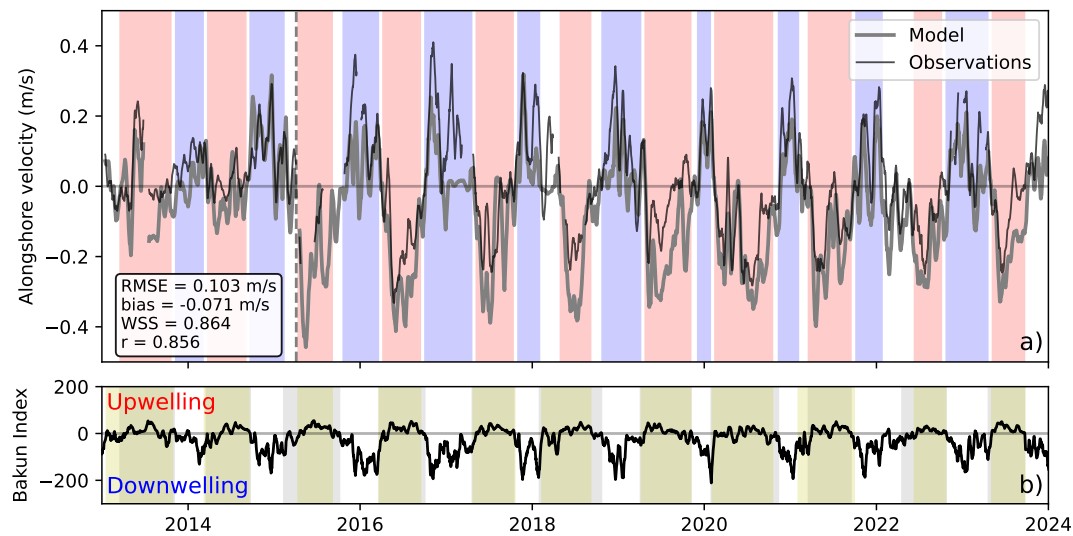

**Figure 2.** Date selection for upwelling (red) and downwelling (blue) periods between 2013 and 2024. Modelled and observed alongshore
velocity (a) at station A1 before April of 2015 (vertical black dashed line) and CE07 after, smoothed using a 15 day running mean as in
Foreman et al. (2011). White areas between upwelling and downwelling indicate periods of transition between the two regimes. The Bakun
index (b) estimates the strength and direction of vertical Ekman transport near the coast, with a positive Bakun index meaning upward
transport (ie. upwelling). Periods of upwelling as defined by the Bakun index (grey) correlate well with those outlined in Hourston and
Thomson (2024) (yellow).



The Bakun Index is based on local wind stress (Bakun, 1973); however, upwelling timing in the Northern California Current System may be influenced by remote winds as far south as $36°$N (Engida et al., 2016), where upwelling favourable winds occur year round (Bakun, 1973). To test the sensitivity of results to the Bakun Index latitude, date selection was repeated using the Bakun index at $45°$N. In general the dates were in close agreement, with upwelling and downwelling dates differing by a week or less in most cases. However, in 2013, upwelling timing differed significantly, starting 18 days later and ending 33 days earlier when using the Bakun Index at $45°$N instead of $48°$N. This discrepancy was unexpected, as upwelling further south typically begins earlier and lasts longer than at higher latitude. Due to this inconsistency and the otherwise similar results between the two Bakun index latitudes, the Bakun index at $48°$N (along with the Hourston and Thomson (2024) dates and the modelled and observed alongshore velocity at A1 and CE07) were deemed a reasonable choice for defining seasonal upwelling and downwelling periods in this region.

### 3.5  Attribution of Interannual Variability Drivers

The volume flux of a property ($PJ$) into JdF is calculated as the sum of contributions from each water mass ($i$):

$$PJ = \sum P_i J_i \tag{1}$$

Here, $P$ represents the mean property value of JdF inflow, and $J \, (\mathrm{m^3})$ is the total volume of water over the analysis period (i.e., the volumetric flow rate ($\mathrm{m^3 \, s^{-1}}$) multiplied by the period length (s), $J = Qt$). For each water mass $i$, $P_i$ and $J_i$ denote the mean property value and the volume contribution, respectively. To analyze the drivers of changes in the flux of properties into JdF ($\Delta PJ$) over time, variations in annual volume and properties are decomposed into components driven by changes in dynamics ($\Delta J_i = J_{i,year} - J_{i,baseline}$), changes in properties ($\Delta P_i = P_i^{year} - P_i^{base}$), and correlated changes ($\Delta P_i \Delta J_i$):

$$\Delta PJ = \sum (\Delta P_i J_i^{base} + P_i^{base} \Delta J_i + \Delta P_i \Delta J_i) \tag{2}$$

Baseline values ($J_i^{base}$ and $P_i^{base}$) are defined as the mean volume flux and mean property value of each water mass, computed for periods of upwelling, downwelling, or the combined downwelling and subsequent upwelling season over the ten years of analysis. Annual values ($J_i^{year}$ and $P_i^{year}$) reported in this paper are computed for a combined downwelling, spring transition, upwelling, and fall transition.

## 4  Results

### 4.1  Salish Sea Inflow

In general, loop water (return flow from JdF) accounts for the most JdF inflow in a given year at 30.7% of inflow or averaging $(4.3 \pm 0.4) \times 10^8 \, \mathrm{m^3}$ (Fig. 3a); the volume flux of loop water remains relatively consistent throughout the year, though it accounts for a larger portion of inflow during downwelling (Fig. 4b). Loop water is made up of a varying mixture of the Pacific sources and of river discharge originating in the Salish Sea. As such, while its contribution to inflow volume and interannual



property and dynamical variability is important (Supplement S3), it should not be treated as an unique or separable source from the other analyzed water masses. Instead, the dynamics and properties of loop water should be thought of as another product of JdF inflow variability partially driven by the Pacific sources. The remainder of this paper will focus on the Pacific sources
of flow into JdF (i.e. parcels originating from one of the outer, pink, boundaries in figure 1).

The water reaching JdF from the Pacific largely originates between potential density surfaces ($\sigma_0$) of 25.4-26.5 $\mathrm{kg\,m^{-3}}$ (based on the first and third transport-weighted quartiles of JdF inflow). The CUC is the largest Pacific source, with $(2.8 \pm 0.4) \times 10^8\,\mathrm{m^3}$ of inflow annually (Fig. 3a) and a mean $\sigma_0$ of 26.5 $\mathrm{kg\,m^{-3}}$, contributing significantly to JdF inflow throughout the year (Fig. 4e). Offshore deep water is predominantly an upwelling source (Fig. 4f), with $(2.1 \pm 0.4) \times 10^8\,\mathrm{m^3}$ annually and a mean $\sigma_0$ of 26.4
$\mathrm{kg\,m^{-3}}$. Offshore surface water enters JdF in small amounts year round (Fig. 4g), totalling $(0.6 \pm 0.2) \times 10^8\,\mathrm{m^3}$ annually with a mean $\sigma_0$ of 25.3 $\mathrm{kg\,m^{-3}}$. North water reaches JdF almost exclusively during upwelling (Fig. 4h), with $(1.5 \pm 0.2) \times 10^8\,\mathrm{m^3}$ annually on a mean $\sigma_0$ of 25.8 $\mathrm{kg\,m^{-3}}$, while south brackish water reaches JdF almost exclusively during downwelling (Fig. 4d), with $(0.9 \pm 0.2) \times 10^8\,\mathrm{m^3}$ annually along a much shallower mean $\sigma_0$ of 23.0 $\mathrm{kg\,m^{-3}}$. South shelf water is predominantly a downwelling source, with $(1.8 \pm 0.4) \times 10^8\,\mathrm{m^3}$ and a mean $\sigma_0$ of 25.2 $\mathrm{kg\,m^{-3}}$; of the six water masses it had the highest
standard deviation in annual inflow.

The length (Fig. 2b and 3b) and strength of upwelling and downwelling periods results in significant year-to-year differences in the water entering the Salish Sea (Fig. 3). The inflows of south shelf and brackish water have a significant (significance level ($\alpha$)=0.05) correlation with the difference in length of consecutive upwelling and downwelling periods (Fig. 3b), the correlation coefficient $(r) = -0.81\,(p = 0.005)$ and $-0.75\,(p = 0.01)$, respectively, while north shelf water is significantly correlated with
the strength of upwelling, r=0.72 (p=0.02). For instance, in 2017, the downwelling season lasted longer than upwelling and the inflow of south shelf and brackish water was higher than typical. In 2020, despite upwelling lasting for a significant amount of time north water inflow volumes were low, owing potentially to the relative weakness of this upwelling period (Fig. 2b). The interannual variability in season length and water mass contribution also manifests in the JdF inflow properties (Fig. 3c-g). The mean $S_A$, DO, and $NO_3$ of JdF inflow are significantly correlated to the difference in upwelling and downwelling length, with
r=0.68 (p=0.03), $-0.72$ (p=0.02), and 0.66 (p=0.04), respectively.





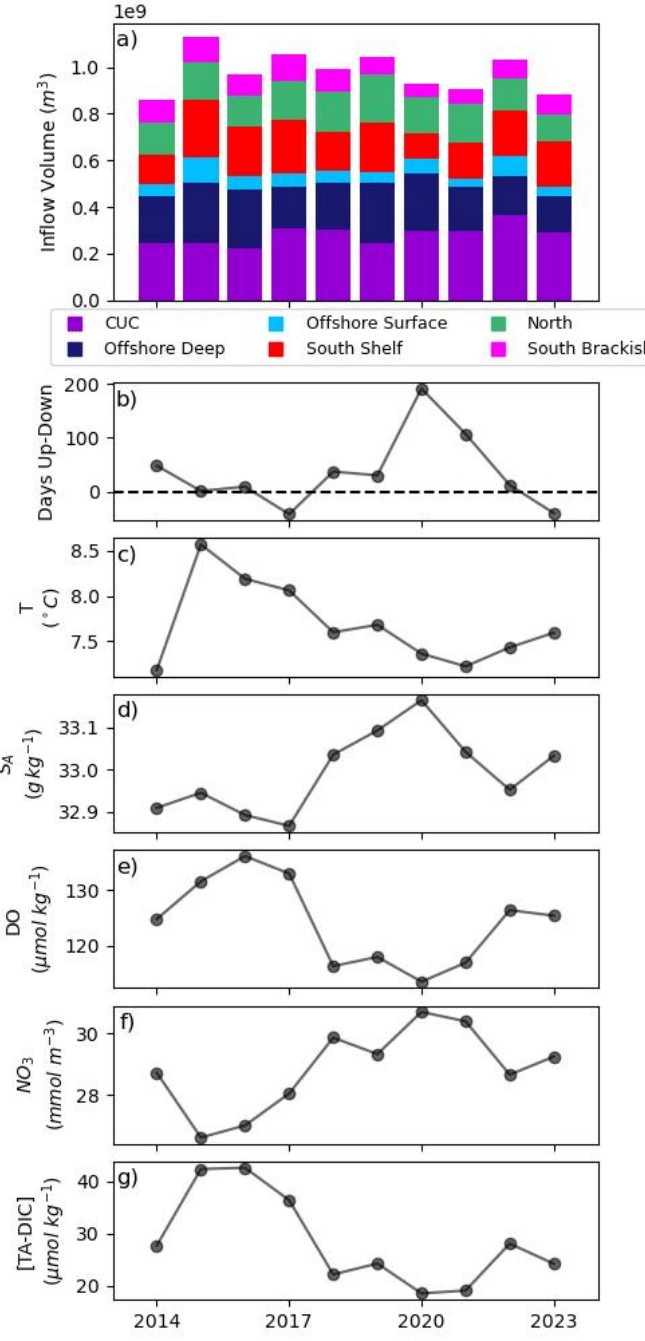

**Figure 3.** (a) Volume from the CUC (purple), offshore deep (navy), offshore surface (light blue), north (green), south (red), and brackish (pink) water into the Salish Sea over one year (combined periods of downwelling, spring transition, upwelling, and fall transition). (b) Difference in the length of upwelling and downwelling in each year. Variability in the transport weighted mean temperature (c), salinity (d), DO (e), $NO_3$ (f), and [TA-DIC] (g) at the mouth of JdF may relate to the flux from each water mass. A version of (a) including the contribution of loop water is included in Supplement S3.



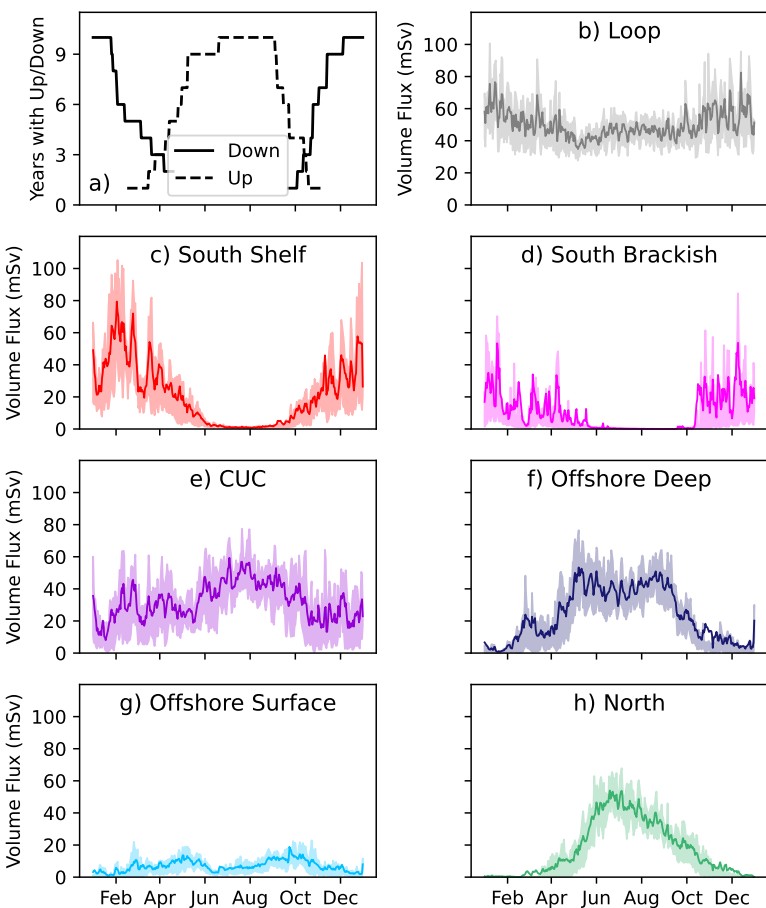

**Figure 4.** Days (a) with upwelling (black, dashed line) or downwelling (black, solid line) are summed over each upwelling and downwelling period to show the common dates over the 10 years of analysis. Daily average transport (1 mSv = $10^3 \, \mathrm{m^3 s^{-1}}$ = 8.64x$10^7 \, \mathrm{m^3 d^{-1}}$) from each water mass averaged (b-h) for the same year-day (solid line) with the inter-quartile ranges about the mean shown in a lighter shade.





## 4.2 Water Mass Properties

### 4.2.1 Observed Properties

The separation of observations into water masses highlights distinct differences between deep (offshore deep and CUC water) and shallow water masses (north and south shelf water, and offshore surface water; Fig. 5). Deep water masses are colder,
saltier, and denser than shallow waters. They are also richer in nutrients, DIC, and TA, while being lower in DO and $CO_3^{2-}$ concentrations. Denitrification (negative N*) is more prevalent than nitrification across all water masses, thus the mean N* are all negative, ranging from -1.8 in offshore deep to -3.8 in the south shelf water (Table A2). In Fig. 5, a high N* refers to a less negative N*, and it is associated with the deep water masses, consistent with their higher concentrations of $NO_3$ and DIP and lower concentrations of the denitrification precursor ($NH_4$).

The two deep water masses (CUC and Offshore Deep) are surprisingly similar, differences in their properties are often not statistically significant (based on a 95% confidence interval p-test, Table A2) or, where they are statistically significant due to abundant observations, are not practically different (as indicated by a Cohen's d <0.5 (Cohen, 1988), suggesting small effect size). Notable exceptions are that the CUC is slightly less dense, spicier, and richer in $NO_2$ than offshore deep water. Shallow water masses differ more-so: offshore surface water is more oxygen rich and has significantly higher aragonite and calcite
saturation than shelf water masses. Among shelf waters, north shelf water is less spicy, while south shelf water exhibits a higher TA.

Seasonal differences are more pronounced in shallow water masses, while deep water masses remain relatively consistent across upwelling and downwelling periods (Fig. 5). These seasonal variations also reveal greater distinctions between shallow water masses. For example, $NO_3$ and DIP concentrations are higher in south shelf water in the summer due to upwelling but
higher in offshore surface water during downwelling due to winter mixing, with little variation in north shelf water (Fig. 5). Offshore surface water shows higher salinity during coastal downwelling, contrasting with upwelling signals in north and south waters.

Trace metal observations are not present or scarce (3 summer observations in the CUC, and none in the winter), with the exception of the offshore water masses (Table A1). While keeping the limited number of CUC trace metal observations in mind,
available data suggest that the CUC has higher concentrations of Co, Fe, and Mn than offshore deep water, even though these water masses are otherwise similar (Fig. 5). Offshore surface water generally has lower trace metal concentrations compared to offshore deep water, except for Mn.

### 4.2.2 Modelled Properties

Given the inconsistent sampling frequency and locations, interannual variability in water mass biogeochemistry is difficult
to assess from observations alone. To Supplement this analysis, modelled variability in water mass properties (Fig. 6) was evaluated. Comparisons between model and observational data (Supplement S1) confirm that the model accurately represents the range and variability of properties. Across the domain, modelled salinity, temperature, DO, $NO_3$, TA, and DIC show high skill (Willmott Skill Scores (WSSs) >= 0.90, Willmott (1981)). Subregion (as defined in Table 1, observations) evaluations





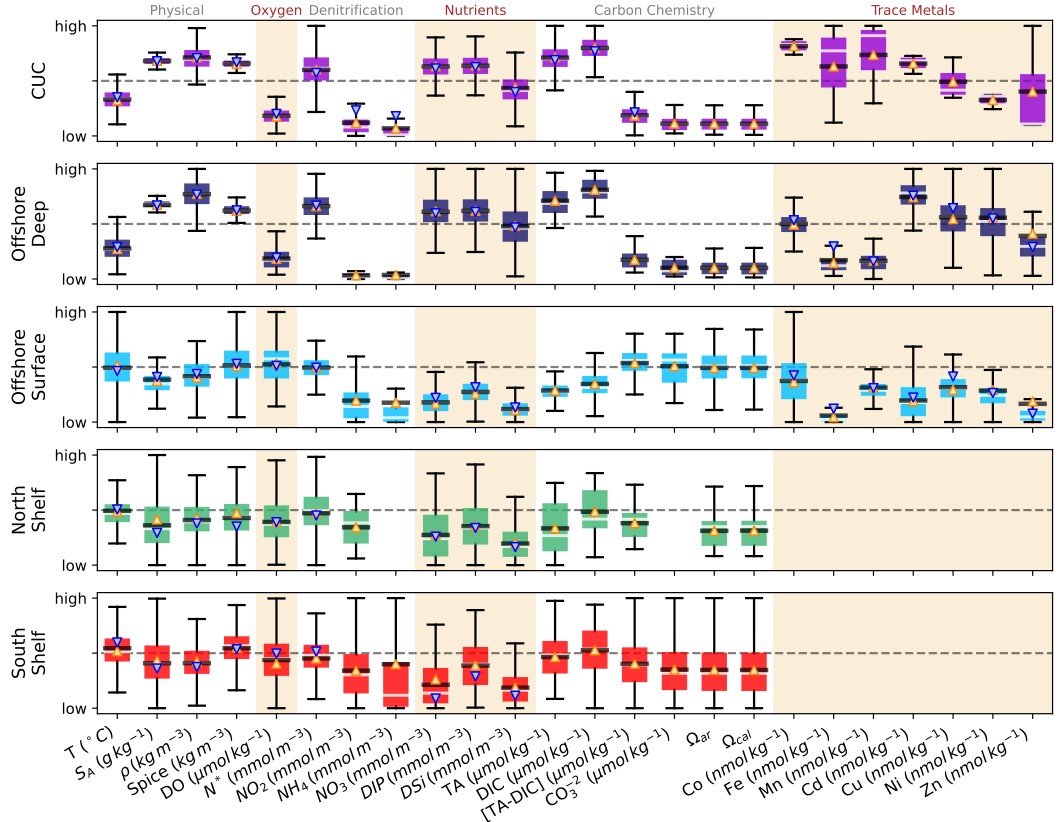

**Figure 5.** Relative property definitions of each water mass based on observations. High and low limits are relative to the other water masses, based on the maximum and minimum whisker values among the five water masses in each property. The light grey line within each box indicates the median value, the upward pointing orange triangle represents the mean over upwelling periods, and the downward pointing blue triangle the mean over downwelling periods (where downwelling observations were available, Table A1). The observed tracer mean and standard deviation in each water mass, and the significance of differences between water masses, is summarized in Table A2.

yield slightly lower skill for some tracers, notably TA in offshore surface water (WSS = 0.76), but all other tracers in all
subregions achieve WSSs >= 0.81.

In general, shallow water masses (south shelf, loop, offshore surface, and north water) exhibit more interannual variability over the analysis period than deep water masses (CUC and offshore deep water, Fig. 6). Temperature change in particular seems to manifest in the shallow water masses, with the elevated water temperatures in the northeast Pacific during the 2014-2016 "Blob" clearly present in the south shelf, offshore surface, and loop water (Fig. 6b). South water has among the most
interannual variability in each water property, with north shelf water only exceeding it in $NO_3$ content. Shallow water masses generally have stronger seasonal cycles than the deep water masses, exhibiting predictable property changes during upwelling:



increased salinity, cooler temperatures, lower DO, and higher concentrations of $NO_3$, TA, and DIC compared to downwelling periods.

Brackish south water, excluded from Fig. 6 due to its distinct property range, is significantly lower in salinity, $NO_3$, DIC, and TA, and higher in DO than the other water masses. It also appears to be even more impacted by the Blob than the other shallow water masses, but otherwise overlaps in temperature with the shallow water masses. Focusing only on downwelling periods due to the lack of brackish south water during most upwelling periods, this water mass has little interannual variability except in temperature.

### 4.3 Drivers of Variability

Using equation 2, we separated the effects of annual variability in the volume of water contributed by each Pacific water mass (Figs. 3a,b) from variability in the properties of those water masses (Figs. 5,6). The cross term (third term in equation 2) is small relative to the impacts of dynamical and property variability, and so is neglected in this analysis. Overall, dynamical variability accounts for the majority of variability in each tracer (Fig. 7). However, property variability plays a notable role in explaining changes in temperature, DO, and $NO_3$ ($\sim \frac{1}{6}$), and a major role in the variability of [TA-DIC] ($\sim \frac{1}{4}$). Notably, in two of the analyzed years (2016 and 2018), property variability plays a larger role than dynamical variability in the change in [TA-DIC] flux.

The water masses that explain the most interannual variability differ from tracer to tracer, and, accordingly, don't correspond with which water masses contribute most to JdF inflow. Offshore deep water and CUC are the two largest Pacific contributors to Salish inflow (Fig. 3), but with the exception of $NO_3$ variability, the contribution of CUC and offshore water to interannual variability is smaller than their volume contribution. The deep water masses contribute similar amounts overall to variability, predominantly in the form of dynamical variability (Fig. 7). Despite south shelf water contributing a smaller portion of JdF inflow annually (Fig. 3) compared to the CUC and offshore deep water, it is the largest driver of interannual variability in all but $NO_3$, where deep waters are more important (Fig. 7). South shelf water contributes to interannual variability in the form of both dynamical and property variability in large amounts, but dynamical variability plays a larger role. The other shallow water masses (north shelf, brackish south, and offshore surface water) are large drivers of interannual variability in DO and [TA-DIC], with their contributions to variability far exceeding their contributions to inflow volume in those tracers. It should be noted, however, that these contributions to variability are themselves variable: the standard deviation in each water mass's contribution exceeds one third of its mean ($\sigma > \frac{1}{3}\bar{x}$), with the CUC and south brackish water showing the largest year-to-year fluctuations.

These variability attributions can be expanded to the inner basins of the Salish Sea using the Pacific connection results from Beutel and Allen (2024). A greater proportion of parcels from intermediate-depth inflow through JdF (south shelf, north shelf, offshore surface waters) reach the Haro Region, compared to deep (CUC, offshore deep) and especially compared to surface (south brackish) sources. As such, the water masses that dominate the variability differ from those through JdF (Fig. 7). South shelf water remains the most important for all but $NO_3$ (where offshore deep water is the most important), but surpasses the importance of CUC in that tracer, south brackish water contributes the least to variability in each tracer, and north shelf water



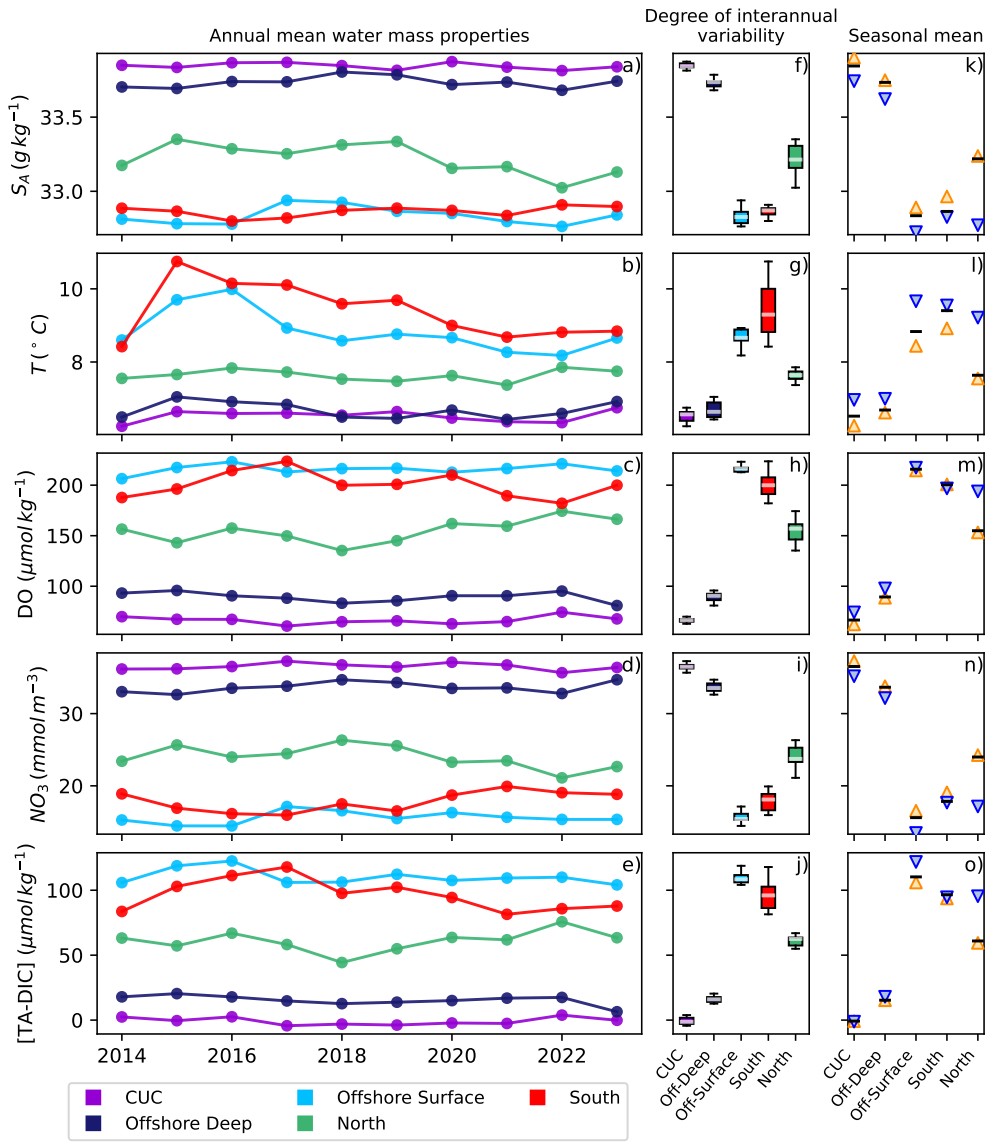

**Figure 6.** Modelled properties and their interannual (a-j) and seasonal (k-o) variability. Transport weighted annual mean (a-e) salinity, temperature, DO, $NO_3$, and [TA-DIC] at the outer boundaries for the CUC (navy), offshore deep (purple), offshore surface (light blue), north (green), and south (red) water masses. Box and whisker plots (f-j) of the combined annual data show the range in properties exhibited by each water mass. Transport weighted mean upwelling (upward pointing orange triangle) and downwelling (downward pointing blue triangle) properties (k-o) compared to the annual mean (black horizontal line) show the seasonal difference in water mass properties in each water mass.

becomes the second or third most important contributor in each tracer. In Puget Sound, approximately twice the percentage of



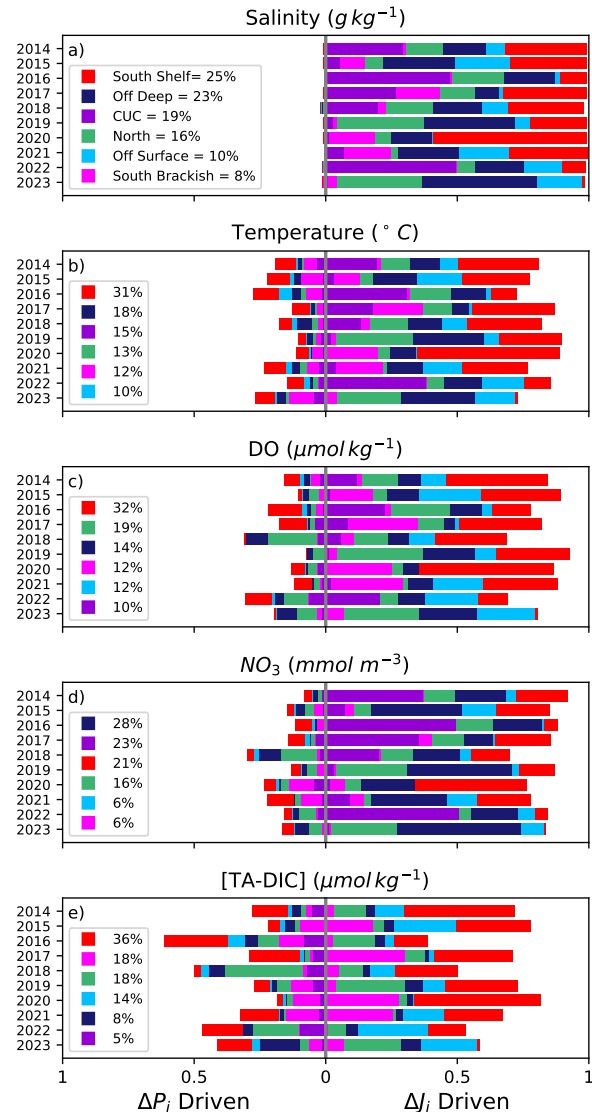

**Figure 7.** Attribution of changes in Salish Sea inflow flux of salinity (a), temperature (b), DO (c), $NO_3$ (d), and [TA-DIC] (e) to interannual differences in water mass inflow volumes (right side of the graph) or to interannual differences in water mass properties (left side). The legend in each figure displays the average percentage contribution of each water mass to interannual variability in the tracer over the ten years of analysis, the water masses are reordered in each legend according to their contribution (top to bottom). A version of this figure including the contribution of loop water is included in Supplement S3.

intermediate water parcels reach the region than deep or surface water parcels. This translates to a similar increase in influence of south and north shelf waters, with south shelf water becoming the dominant contributor to variability for all tracers, including





NO$_3$. Although deep water masses contribute less to variability in the inner basins than in JdF inflow, they remain significant sources of variability in salinity and NO$_3$.

## 5 Discussion

Modelled and observed properties of water masses entering the Salish Sea through JdF highlight key drivers of interannual variability in inflow properties. Variability in the volume contributions of different water masses generally plays a larger role than variability in their properties; however, property variability remains significant for temperature, DO, NO$_3$, and the OA proxy [TA-DIC]. South shelf and deep water masses are the largest drivers overall to interannual variability in the flux of tracers through JdF inflow, with shallow water masses tending to be more important for DO and [TA-DIC] variability and deep water masses tending to be more important for NO$_3$ variability. Combining observed water mass properties with model results can help reveal drivers of variability in biogeochemical tracers not explicitly represented in the model.

### 5.1 Drivers of Biogeochemical Variability

#### 5.1.1 Physical Properties

Although both are physical properties and conservative tracers, salinity and temperature differ markedly in their interannual trends (Fig.6a,d) and underlying drivers (Fig.7a,b). Salinity variability is almost entirely explained by dynamics, whereas temperature variability arises from both dynamic and property variations. Shallow water masses exhibit substantial interannual and seasonal temperature variability (Fig. 6b,g,l), whereas salinity remains relatively constant within each water mass. Salinity drives density variability in this region (Broatch and MacCready, 2022); since the water masses were defined primarily by density-related criteria (salinity, depth), it is unsurprising they show minimal interannual salinity change.

Temperature however, can vary within a water mass without substantially altering its density, thus allowing notable temperature shifts independent of water mass classification. During warmer inflow in Blob years (2015, 2016, and 2017, Fig.6) temperature variability was predominantly driven by shallow water masses—particularly south shelf and brackish waters (Fig.7b). Cooler-than-average inflow years (2014 and 2021; Fig. 6) similarly underscore the influence of these shallow southern water masses on inflow temperature.

South shelf water, the primary contributor to temperature variability in JdF inflow, originates from the southern end of the CCS (Checkley and Barth, 2009) where sea surface temperature (SST) has increased by $\sim 2.5°$ C in the past 70 years with strong decadal and multi-decal variability therein (Lund, 2024). Offshore deep water, the second-largest contributor to temperature variability, is defined in this study (Table1) within the deeper portions of the CC (Auad et al., 2011), which has also experienced significant anthropogenic warming (Field et al., 2006). The CUC, another large contributor to temperature variability, is getting spicier (Meinvielle and Johnson, 2013; Maier et al., 2025). Although the CUC currently influences JdF inflow primarily via dynamical variability, continued northward shifts in its properties (Meinvielle and Johnson, 2013) may increase the importance of its property-driven impacts over longer timescales. The isopycnals upwelled into JdF from the CUC





and offshore deep water closely align with those found in Maier et al. (2025), where the 26.4 and 26.5 $\mathrm{kg\,m^{-3}}$, and to a lesser extent the 26.6 $\mathrm{kg\,m^{-3}}$, isopycnals were regularly upwelled onto the shelf. Maier et al. (2025) found that the transport of these isopycnals onto the shelf had no significant correlation with upwelling and downwelling strength or timing, aligning with the lack of correlation found between CUC transport and upwelling metrics in this study. Given the similar present properties of the CUC and offshore deep water (Fig.5), an important open question is whether these two water masses will remain aligned in the

future. This question is especially relevant given their substantial contributions to both inflow volume and property variability into JdF (Figs.3, 7).

### 5.1.2  Oxygen and Nutrients

DO and $NO_3$ variability are both predominantly driven by dynamical changes in water mass inflow, though property variability also plays a notable role, explaining roughly one-sixth of total variability. Although DO and $NO_3$ are related, with elevated

nutrient concentrations commonly associated with lower oxygen conditions, the water masses driving variability in these two tracers differ substantially. Variability in DO is primarily controlled by shallow water masses, dominated by south shelf water and followed by north shelf water, both of which display significant interannual variability in volume (Fig.3) and oxygen content (Figs.6,5). In contrast, $NO_3$ variability is predominantly influenced by deep water masses, which exhibit minimal interannual variability in $NO_3$ properties (Figs.6,5).

Deep waters almost entirely drive $NO_3$ variability through their dynamical contributions. This dependence is consistent with observational data, which show little variability in $NO_3$ and DIP concentrations within deep water masses across the entire dataset (Fig. 5). DSi, however, exhibits greater observed variability and relatively lower concentrations compared to $NO_3$ and DIP. Given these differences, the flux of DIP into JdF may follow a similar variability pattern to $NO_3$, whereas DSi variability likely depends on additional factors. It is plausible that deep water masses still drive much of the DSi flux variability but that

increased property variability also plays a role, or its possible that shallow waters have a greater contribution due to dissolved silica inputs from freshwater (DeMaster, 1981).

Concurrent with increasing spice, nutrient concentrations in the CUC have risen while DO concentrations have declined near JdF, driven by an enhanced contribution of PEW (Meinvielle and Johnson, 2013). The associated shoaling and intensification of the CUC (Meinvielle and Johnson, 2013; Bograd et al., 2008) may further elevate the flux of nutrient-rich, low-oxygen water

into the JdF inflow. In the southern end of the CCS, $NO_3$ flux to the shelf has nearly doubled from 1980 to 2010 due largely to increased wind-driven upwelling intensity (Jacox et al., 2015), this intensification has also been observed in the northern end of the system (Foreman et al., 2011) and impact south shelf water nutrient content both remotely and locally. Meanwhile, the rising SST in the region (Lund, 2024) is increasing stratification (Bograd and Lynn, 2003), potentially limiting mixing between south shelf and CUC waters, but has a smaller impact than wind-driven increases in upwelling (Jacox et al., 2015). This increasing

SST also directly lowers oxygen concentrations in shallow waters through reduced oxygen solubility and elevated respiration rates (Gruber, 2011). Additionally, the North Pacific Current, feeding the California Current (and thus offshore deep water), is currently undergoing deoxygenation linked to diminished ventilation from upstream stratification (Mecking and Drushka, 2024; Bograd et al., 2008), further contributing to a potential decreasing oxygen trend in JdF inflow.





### 5.1.3 Carbonate Chemistry Constituents

Among the tracers examined, [TA-DIC] is the most strongly influenced by property variability, which explains approximately one-quarter of its interannual variability (Fig. 7). Shallow waters dominate this variability more than for any other tracer: all four shallow water masses contribute more to interannual changes in [TA-DIC] than the deep water masses. When TA and DIC are analyzed separately (not shown), their interannual variability closely mirrors that of salinity, consistent with previously reported covariance in the region (Fry et al., 2015). This covariance supports the use of [TA-DIC] as a more informative metric
for diagnosing drivers of OA in coastal systems (Xue and Cai, 2020).

The strong influence of shallow waters on [TA-DIC] variability is consistent with current understanding of the carbonate system in the Northeast Pacific. Deep waters in this region are largely isolated from anthropogenic $CO_2$, but naturally exhibit high DIC concentrations due to the long-term accumulation of $CO_2$ from microbial respiration (Feely et al., 2004). In contrast, shallow waters —historically lower in DIC— have shown a significant upward trend in DIC over the past 30 years, attributed
to anthropogenic $CO_2$ uptake (Franco et al., 2021). The prominent role of shallow waters in driving [TA-DIC] variability in JdF inflow underscores the vulnerability of the Salish Sea to acidification via anthropogenic $CO_2$ uptake, as stressed in previous Salish Sea OA research (e.g., Jarnikova et al., 2022; Feely et al., 2010).

### 5.1.4 Denitrification

Our analysis of denitrification is limited to water mass differences discerned from observations, as model outputs do not
include $N^*$, $NO_2$, or $NH_4$. Based on mean observed $N^*$ values (Table A2), denitrification appears to be active in all water masses, but is 1.5 to 2 times as strong in the shallow water masses compared to the deep. This result is somewhat unexpected, as we anticipated stronger denitrification in deep waters due to their lower oxygen concentrations (Casciotti et al., 2024). One possible explanation is that shallow waters have more contact with sediment where denitrification is occurring (Devol, 2015), and that this signal is carried in the shallow water masses despite these waters existing above the typical oxygen minimum
zone depth in the region (Pierce et al., 2012).

Predicting temporal variability in denitrification and nitrification is challenging due to the number of interacting tracers involved. The strength of denitrification depends on the availability of $NO_3$, DIP, Fe, Cu, organic matter, and DO (Casciotti et al., 2024; Li et al., 2015). In general, $NO_3$ and DO are negatively correlated due to respiration, and as such, variability in denitrification may be expected to follow similar patterns to these two tracers (Figs. 7c,d), with both property and dynamical
variability playing important roles. However, the drivers differ: DO variability is dominated by changes in shallow water masses, while $NO_3$ (and likely DIP) is primarily influenced by the dynamics of deep water masses. The combined effects of decreasing oxygen and shifting nutrient distributions (Sec. 5.1.2) make it difficult to predict how denitrification in shallow waters will evolve given its sensitivity to multiple interacting drivers.



### 5.1.5 Trace Metals

The resolution of trace metal observations in the study region is unfortunately insufficient to attribute distinct trace metal properties to each water mass. However, observations from within the Salish Sea and other parts of the CCS provide useful benchmarks for interpreting the trace metal concentrations observed in offshore and CUC water masses. For example, at a surface depth of 75 m approximately 150 km offshore of San Francisco, Bruland (1980) reported concentrations of Cd = $0.29\,\mathrm{nmol\,kg^{-1}}$, Cu = $1.35\,\mathrm{nmol\,kg^{-1}}$, Ni = $4.34\,\mathrm{nmol\,kg^{-1}}$, and Zn = $1.27\,\mathrm{nmol\,kg^{-1}}$. With the exception of Zn, these values

closely match the offshore surface concentrations from GEOTRACES data used in this study (Table A2). At 490 m in the same location, Bruland found Cd = $1.04\,\mathrm{nmol\,kg^{-1}}$, Cu = $1.53\,\mathrm{nmol\,kg^{-1}}$, Ni = $7.19\,\mathrm{nmol\,kg^{-1}}$, and Zn = $5.37\,\mathrm{nmol\,kg^{-1}}$—all within one standard deviation of our offshore deep water measurements. Additional measurements of Fe and Mn at 250 m reported concentrations of $1.31\,\mathrm{nmol\,kg^{-1}}$, $1.73\,\mathrm{nmol\,kg^{-1}}$, respectively (Landing and Bruland, 1987), both comparable to the mean values for CUC water in this study. It is important to note, however, that standard deviations for Zn and Ni are relatively

large in this study (Table A2), which limits the precision of water mass-specific comparisons.

Within the Salish Sea studies of Cd and Cu concentrations have highlighted Pacific inflow through JdF as the largest contributor of these trace metals (Kuang et al., 2022; Waugh et al., 2022). At JdF stations Cd and Cu were $0.8\,\mathrm{nmol\,kg^{-1}}$ and $1.49\,\mathrm{nmol\,kg^{-1}}$ in the inflowing layer, respectively (Kuang et al., 2022; Waugh et al., 2022). These values closely match mean concentrations observed in offshore deep and CUC water masses (Table A2), suggesting that they could be sources of these

trace metals into JdF inflow. However, without trace metal measurements of shelf waters, the degree of their contribution cannot be quantified.

The tracer metal characteristics of the CUC, offshore deep, and offshore surface water reveal distinctions in the water masses not evident from the other biogeochemical tracers. The only significant difference between the trace metal concentrations of the CUC with the offshore water masses (Table A2) is its elevated concentrations of lithogenic metals —Co, Fe, and Mn (Lam

et al., 2015; Zheng and Sohrin, 2019)— likely reflecting its path along the continental slope and the influence of bedrock weathering. In contrast, the offshore water masses differ from each other in all measured trace metals, with offshore deep water showing higher concentrations than offshore surface water of both lithogenic and anthropogenic (Cu, Cd, Ni, Zn; Luo et al. (2022)) metals, with the exception of Mn (Table A2). Although atmospheric deposition may be a major source of anthropogenic and lithogenic metals to the surface ocean in the Northeast Pacific (Mangahas et al., 2025), trace metals in the surface layer

may be depleted due to biological uptake (particularly Fe and Zn, to a lesser extent Mn, Ni, and Cu, and trace amounts of Cd; Twining and Baines (2013)) or particle reactivity (Fe, Mn, Zn, and Cd; Bruland et al. (1994)).

### 5.2 Limitations

#### 5.2.1 Observations

As is particularly evident in the case of trace metals, observations of biogeochemical conditions remain sparse for certain

tracers, water masses, and time periods. For example, the north water mass draws from a relatively small spatial domain (Table 1), due to the proximity of Juan de Fuca Strait to the northern boundary of the CCS. Consequently, this water mass is





less well-sampled than others and lacks any observations of $NH_4$ or $CO_3^{2-}$ (Table A1). Low sample counts, whether in specific water masses or for particular tracers, reduce the statistical significance of some of the mean properties estimated in this study (Table A2).

Unsurprisingly, the observational dataset is heavily biased toward the summer months/upwelling periods across all water masses (Table A1). Measurements of carbon chemistry components are particularly seasonal, with DIC and TA during down-welling only available for the CUC. This seasonal bias likely affects mean conditions reported in the study for variables with strong seasonal cycles. For instance, DIC is drawn down during summer by surface-layer primary production (Moore-Maley et al., 2016), which may result in an underestimation of DIC concentrations in shallow water masses due to summer-skewed
sampling.

       To increase the number of measurements available, observations were collated from outside the LiveOcean domain. The water mass definitions are based on previous research on the northern CCS (Checkley and Barth, 2009; Thomson and Krassovski, 2010) and the analysis of water mass properties dependent on those definitions, such as the decision to split offshore water into a surface and deep components due to the presence of distinct $NO_3$ groups (Supplement S2.2). The mean water mass properties
are sensitive to these definitions (Supplement S2). Additionally, since the observations do not spatially coincide with the model analysis domain (Fig. 1), they are not directly comparable with the Lagrangian model output. In this paper, observed water mass properties are thus used qualitatively—for instance, to compare whether the CUC is nutrient-rich relative to south shelf water—rather than as definitive values.

### 5.2.2  Model

The transport estimates in this study are based on parcels that complete a full trajectory—from initialization either to the shelf or offshore boundary, or returning to the initialization section. Water that is already within the analysis domain (e.g., "lost" parcels, representing ~1% of seeded parcels) or tidally pumped water (parcels that briefly exit and return across the initialization section within two tidal cycles) are excluded from these estimates. As a result, the reported volumes should not be interpreted as estimates of total transport through JdF; doing so would result in a substantial underestimate (Beutel and Allen,
510  2024).

       As noted in Section 3.2, Ariane does not include sub-grid scale mixing, and instead relies on mixing resolved by the underlying numerical model. Sub-grid scale mixing (e.g., as random walk steps) would generally act to slow parcel trajectories, potentially increasing parcel loss rates. However, prior Lagrangian work in the Salish Sea using a model with 500 m resolution found the effect of sub-grid mixing on estuarine exchange to be minimal—slowing parcels by no more than 2% even in a
region of intense vertical mixing (Allen et al., 2025). In that case, the model captured the largest eddies, which contribute most to mixing; as a result, omitting sub-grid scale mixing had limited impact. Although the LiveOcean model used here has coarser resolution, the study region exhibits much weaker mixing than the site assessed in Allen et al. (2025). Therefore, large-scale eddies are still expected to be well resolved, and the omission of sub-grid scale mixing likely has a similarly small effect on the results.



The 10-year model period analyzed in this study is not sufficient to resolve long-term trends in biogeochemical conditions or multiple transitions of low-frequency climate modes such as the PDO and NPGO (Di Lorenzo et al., 2008). Observed changes in CCS water properties—driven by both natural variability and anthropogenic forcing—are well documented and expected to continue (Bograd et al., 2019; Meinvielle and Johnson, 2013; Di Lorenzo et al., 2008). Applying the decomposition framework (Equation 2) to a longer time series, or using a pre-industrial baseline instead of a decadal mean, would likely reveal a larger

role for property variability than was detected here. Additionally, extreme events may be underrepresented in the 10-year model record, as ocean models often struggle to reproduce anomalies. For instance, while model–observation agreement was generally good for DO and $NO_3$ (Supplement S1), the anomalous conditions observed on the shelf in 2019 (low nutrients, high DO) and in 2021 (high nutrients, extreme hypoxia; Franco et al. 2023) are not apparent in either the modelled JdF inflow (Fig.3) or water mass properties (Fig.6).

**6    Conclusions**

Water mass contributions to JdF inflow, and the modelled and observed biogeochemical properties of these water masses, highlight the diverse drivers of interannual variability in tracer flux.

   Deep water masses (CUC, offshore deep) are the dominant Pacific contributors (i.e. excluding recirculated JdF outflow) to annual JdF inflow. These waters are denser, spicier, nutrient-rich, and low in dissolved oxygen compared to shallow water

masses (south shelf, north shelf, offshore surface, south brackish), and they exhibit relatively uniform properties across different deep sources. Their influence on biogeochemical variability is primarily through dynamical variability, especially for $NO_3$, where deep waters are the principal driver.

   In contrast, shallow water masses are more distinct from one another and exhibit greater interannual property variability than the deep waters. South shelf water, for instance, is spicier and more nutrient-poor than north shelf water, and was noticeably

impacted by warming in the Blob years. The volume of inflow from shallow water masses depends on seasonal wind forcing: longer upwelling periods reduce contributions from south shelf and brackish waters, while stronger upwelling increases north shelf water inflow. Despite their smaller overall volume contribution, shallow water masses disproportionately impact variability in tracer fluxes, particularly in [TA–DIC] and DO, due to their variable properties and inflow volumes.

   South shelf water and the deep water masses are the most important drivers of biogeochemical variability in JdF inflow.

As the CUC continues to be influenced by an increasing contribution of PEW, it is likely to play an even larger role in both dynamical and property variability. South shelf water will be affected both by changing CUC properties—via mixing on the shelf—and by direct anthropogenic impacts in the surface ocean, including $CO_2$ uptake and warming. Offshore deep water, likely representing the subsurface portion of the CC, is already undergoing warming and deoxygenation. While CUC and offshore deep water presently have remarkably similar properties, ongoing anthropogenic change may cause them to diverge.

Understanding the vulnerability of JdF inflow to future change will require accurate projections of both circulation and water mass properties in shelf and subsurface currents at the northern end of the CCS.



*Code and data availability.* Simulation setup and results files, and the collated observational dataset are archived at the Federated Research Data Repository, https://doi.org/10.20383/103.01339. Analysis, and figures code are available at the following Github repository: https://github.com/rbeutel/PI_BIOGEO_PAPER.



 **Appendix A: Tables**

**Table A1.** Number of observations in each water mass (Table 1) during upwelling (summer) and downwelling (winter) periods. The source of observations for each variable is summarized in Table 2.

| Variable | Offshore Surface | | Offshore Deep | | CUC | | North | | South | |
|---|---|---|---|---|---|---|---|---|---|---|
| | Summer | Winter | Summer | Winter | Summer | Winter | Summer | Winter | Summer | Winter |
| T | 43,821 | 17,912 | 91,293 | 34,788 | 88,373 | 37,567 | 11,216 | 5,902 | 175,633 | 83,273 |
| $S_A$ | 69,658 | 28,334 | 112,754 | 45,852 | 101,448 | 49,229 | 37,534 | 28,208 | 175,973 | 83,762 |
| DO | 38,122 | 17,117 | 79,491 | 31,053 | 74,869 | 32,389 | 6,594 | 2,308 | 127,663 | 65,400 |
| $NO_3$ | 8,802 | 2,579 | 4,447 | 1,474 | 6,065 | 578 | 5,054 | 587 | 3,030 | 1,225 |
| $NO_2$ | 304 | 0 | 193 | 0 | 908 | 18 | 36 | 0 | 1,714 | 0 |
| $NH_4$ | 144 | 0 | 92 | 0 | 775 | 18 | 0 | 0 | 1,590 | 0 |
| DIP | 8,903 | 2,585 | 4,796 | 1,584 | 6,121 | 592 | 5,173 | 568 | 1,863 | 12 |
| DSi | 8,390 | 2,445 | 4,769 | 1,924 | 5,979 | 509 | 5,037 | 574 | 1,861 | 12 |
| TA | 297 | 0 | 176 | 0 | 1,010 | 89 | 35 | 0 | 1,726 | 0 |
| DIC | 280 | 0 | 173 | 0 | 1,013 | 88 | 36 | 0 | 1,747 | 0 |
| $CO_3^{2-}$ | 54 | 0 | 38 | 0 | 274 | 0 | 0 | 0 | 561 | 0 |
| $\Omega_{ar}$ | 268 | 0 | 163 | 0 | 835 | 0 | 34 | 0 | 1,680 | 0 |
| $\Omega_{cal}$ | 268 | 0 | 163 | 0 | 835 | 0 | 34 | 0 | 1,680 | 0 |
| Cd | 219 | 53 | 152 | 31 | 3 | 0 | 0 | 0 | 0 | 0 |
| Co | 219 | 53 | 152 | 31 | 3 | 0 | 0 | 0 | 0 | 0 |
| Cu | 219 | 53 | 152 | 31 | 3 | 0 | 0 | 0 | 0 | 0 |
| Fe | 219 | 53 | 152 | 31 | 3 | 0 | 0 | 0 | 0 | 0 |
| Mn | 219 | 53 | 152 | 31 | 3 | 0 | 0 | 0 | 0 | 0 |
| Ni | 219 | 53 | 152 | 31 | 3 | 0 | 0 | 0 | 0 | 0 |
| Zn | 218 | 53 | 151 | 31 | 3 | 0 | 0 | 0 | 0 | 0 |




**Table A2.** Mean (standard deviation) of observed water mass properties. Superscripts of the letter 'X' followed by a number denote the water masses with which a given property is not significantly different at the 0.05 confidence interval, the number corresponding to each water mass is provided in the table header.

| Variable | 1.Offshore Deep | 2.CUC | 3.South | 4.North | 5.Offshore Surface |
|---|---|---|---|---|---|
| T $(^{\circ}\mathrm{C})$ | 6.4 (1.1) | 6.9 (1.2) | 9.6 (1.8) | 8.9 (1.7) | 8.9 (3.2) |
| $S_A$ $(\mathrm{g\,kg^{-1}})$ | 34.0 (0.2) | 34.0 (0.2) | 33.0 (0.6) | 32.8 (0.7) | 32.9 (0.6) |
| $\rho$ $(\mathrm{kg\,m^{-3}})$ | 1028.0 (0.7) | 1027.6 (0.7) | 1025.5 (0.8)$^{X4}$ | 1025.5 (0.9)$^{X3}$ | 1025.6 (1.0) |
| Spice $(\mathrm{kg\,m^{-3}})$ | -0.2 (0.1) | -0.1 (0.1) | -0.4 (0.3) | -0.7 (0.4) | -0.4 (0.4) |
| DO $(\mu\mathrm{mol\,kg^{-1}})$ | 84.0 (41.3) | 82.9 (33.3) | 197.7 (78.2) | 180.7 (76.4) | 236.7 (72.2) |
| $N^*$ $(\mathrm{mmol\,m^{-3}})$ | -1.8 (1.5) | -2.4 (2.3) | -3.9 (2.0) | -3.7 (2.8) | -3.5 (1.3) |
| $NO_2$ $(\mathrm{mmol\,m^{-3}})$ | 0.0 (0.0) | 0.1 (0.1) | 0.2 (0.1) | 0.2 (0.1)$^{X5}$ | 0.1 (0.1)$^{X4}$ |
| $NH_4$ $(\mathrm{mmol\,m^{-3}})$ | 0.0 (0.1)$^{X2,X5}$ | 0.1 (0.2)$^{X1,X5}$ | 0.4 (0.8) | | 0.2 (0.4)$^{X1,X2}$ |
| $NO_3$ $(\mathrm{mmol\,m^{-3}})$ | 32.3 (6.9) | 33.4 (5.3) | 11.2 (10.5) | 14.5 (11.0) | 9.5 (5.9) |
| DIP $(\mathrm{mmol\,m^{-3}})$ | 2.4 (0.5) | 2.4 (0.4) | 1.5 (0.8) | 1.3 (0.7) | 1.0 (0.4) |
| DSi $(\mathrm{mmol\,m^{-3}})$ | 57.8 (21.0) | 52.7 (15.8) | 22.4 (16.7) | 23.5 (16.0) | 14.2 (8.6) |
| TA $(\mu\mathrm{mol\,kg^{-1}})$ | 2268.5 (24.0) | 2268.7 (21.3) | 2217.6 (33.1) | 2190.5 (48.0) | 2180.7 (25.8) |
| DIC $(\mu\mathrm{mol\,kg^{-1}})$ | 2239.7 (48.6) | 2234.6 (45.1) | 2102.9 (97.5) | 2083.0 (93.4) | 2016.2 (68.3) |
| $[\mathrm{TA-DIC}]$ $(\mu\mathrm{mol\,kg^{-1}})$ | 29.5 (26.3) | 34.0 (28.7) | 115.0 (71.5) | 106.7 (48.7) | 164.6 (47.4) |
| $CO_3^{2-}$ $(\mu\mathrm{mol\,kg^{-1}})$ | 57.3 (10.5) | 59.0 (10.6) | 98.8 (36.3) | | 124.9 (28.1) |
| $\Omega_{ar}$ | 0.8 (0.2)$^{X2}$ | 0.8 (0.2)$^{X1}$ | 1.5 (0.6)$^{X4}$ | 1.4 (0.4)$^{X3}$ | 1.8 (0.4) |
| $\Omega_{cal}$ | 1.2 (0.3)$^{X2}$ | 1.3 (0.3)$^{X1}$ | 2.3 (1.0)$^{X4}$ | 2.2 (0.6)$^{X3}$ | 2.9 (0.7) |
| Cd $(\mathrm{nmol\,kg^{-1}})$ | 0.8 (0.2)$^{X2}$ | 0.7 (0.1)$^{X1}$ | | | 0.2 (0.2) |
| Co $(\mathrm{nmol\,kg^{-1}})$ | 51.6 (12.0) | 80.0 (5.3) | | | 40.1 (17.9) |
| Cu $(\mathrm{nmol\,kg^{-1}})$ | 1.9 (0.4)$^{X2}$ | 1.8 (0.3)$^{X1,X5}$ | | | 1.4 (0.4)$^{X2}$ |
| Fe $(\mathrm{nmol\,kg^{-1}})$ | 0.6 (0.6) | 2.2 (1.3) | | | 0.2 (0.4) |
| Mn $(\mathrm{nmol\,kg^{-1}})$ | 0.7 (0.3) | 2.2 (0.9) | | | 1.1 (0.4) |
| Ni $(\mathrm{nmol\,kg^{-1}})$ | 6.3 (2.8)$^{X2}$ | 4.9 (0.3)$^{X1,X5}$ | | | 4.7 (2.5)$^{X2}$ |
| Zn $(\mathrm{nmol\,kg^{-1}})$ | 9.0 (8.8)$^{X2}$ | 9.3 (9.7)$^{X1,X5}$ | | | 3.8 (7.9)$^{X2}$ |



*Author contributions.* BB and SEA conceptualized the project, BB planned the methods, performed the analysis, wrote the manuscript draft, and completed data curation under supervision from SEA. JX provided access and insight into the LiveOcean model. MM conceptualized the inclusion of trace metals in the project. SEA, JX, and MM also contributed their expertise to the manuscript in the review and editing stage.

*Competing interests.* The authors declare that they have no conflict of interest.

*Acknowledgements.* The authors are thankful for the opportunity to conduct research on the coast of and about the Salish Sea, the traditional, ancestral, and unceded territory of the Coast Salish peoples. We recognize their enduring presence on these waters and express our gratitude for their stewardship of this territory. This work was funded by the NSERC-CGS D scholarship to BB, and NSERC-Discovery RGPIN-2022-03112 and Compute Canada RRG 2648-RAC 2019 to SEA.



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
