# Peer review of "Water property variability into a semi-enclosed sea dominated by dynamics, modulated by properties."

_EGUsphere, 2025_

## Author Response (AR1)

We would again like to thank the reviewers for their detailed comments, which have enhanced the manuscript. In this document we address reviewers' comments in a point-by-point manner with excerpts from changed portions of the text provided where applicable. Reviewer comments are in black, our responses are in blue.

**Reviewer 1**

In this manuscripts, the authors present a detailed analysis of the role of changes in advection and source water properties in driving variability in water properties within the Salish Sea. Their analysis is based on Lagrangian experiments performed from outputs of an ocean model, and informed by observational data. The authors present interesting results whereby the processes governing variability differ across tracers, and discuss the implications for future changes. The manuscript is convincing, well-structured and well-written. I believe it will be ready for publication following minor revisions.

**General comments**

- 1.1 The authors currently present the results primarily in terms of years and seasons. I suggest that the authors also provide some of the results in terms of density or depth, since we can expect different water masses to contribute differently to different depth-levels. This perspective could for instance help to assess local impacts, since different areas/depths of the Salish Sea are probably fed by specific density-levels. For example, the authors could provide figures similar to Fig. 3a and 7 but plotted against density. These discussion of these results could be linked to densities presented at L260 onward. Thank you for this suggestion, we agree that adding the density perspective enhances the relevance of the results to the interior of the Salish Sea. The revised draft includes an attribution analysis plotted against density ranges (Fig. 9). This analysis shows that the CUC and offshore deep water contribute to variability in higher density ranges, south shelf and offshore surface water in lower density ranges, and north shelf water in all density ranges. These results are placed in the context of JdF inflow by also including a Hovmöller diagram of monthly climatology of density with depth at the initialisation cross-section (Fig. 4a), and histograms of parcel densities (Figs. 4b,c) and depth (Fig. S9) from each of the water sources at the beginning and end of their trajectories.
- 1.2 Related to this point, it would be informative to show whether the density of water entering the JdF line varies interannually and seasonally, and show how this variability relates to changes in upwelling/downwelling or shifts in water source properties. The variability in the density of water entering JdF and the drivers of this density variability were found to match very closely with that of salinity since density is primarily driven by salinity in the region this correlation did not come as a surprise. With this in mind, it was decided that including a variability analysis of both tracers in the main text would be redundant. This choice is now stated explicitly in the text (line 314) and figures of density variability and the drivers of it are be included in the supplement (Figs. S10, S11).

• Line 314: "It should be noted that density varies strongly with salinity in the region (r=0.9, p=0.001); only interannual salinity variability is shown but a density comparison is available in Fig. S10."

The seasonal variability in the density of JdF inflow was studied in Beutel & Allen (2024). During periods of upwelling, inflow originates primarily from the north shelf, offshore deep, and CUC water. During periods of downwelling, inflow originates primarily from the south shelf and south brackish water. This change in seasonal source waters was accompanied by a large difference in the density of water entering JdF on seasonal scales, with the inflow being significantly more dense during periods of upwelling. Inflow to the Salish Sea also is much more likely to reach the inner basins during upwelling, as intermittent winds during downwelling often change the direction of flow in JdF. The seasonal source water and density differences from Beutel & Allen (2024) have been added to the regional description and results when discussing seasonal differences in JdF inflow:

- Line 57: "Poleward wind drives downwelling in the region ... and local water primarily originates from the southern shelf and continental slope. Under strong poleward winds, outflow from the Columbia River ... can also contribute to the inflow (Giddings and MacCready, 2017)."
- Line 61: "During upwelling, water originates from the northern shelf and offshore, generally within the top 300 m of the water column (Beutel and Allen, 2024). The contrasting properties of downwelled (less dense, fresher, more oxygen-rich, and nutrient- and DIC-poor) and upwelled water account for much of the seasonal variability in JdF inflow (Masson, 2006)."
- Line 298: "The timing and dominance of source waters align well with those in Beutel and Allen (2024)."
- Line 396: "On seasonal scales the relative importance of water source to variability shift closer to their volume contribution (Fig. 3; Beutel and Allen (2024))."
- Line 419: "However, Beutel and Allen (2024) also revealed a seasonal difference in the connection of JdF inflow to the inner basins, with parcels being much more likely to reach the inner basins during periods of upwelling."
- 2. I invite the authors to clarify how observational data was used in this study, and how the inter-annual variations in the source water properties were computed. If I understand correctly, these inter-annual variations were derived from the model, but this should be stated explicitly in the method section, along with details on the procedure of how they were calculated (e.g. temporal averaging, selection of grid cells, etc.). Stating clearly how the observations will be used in section 3.3 should help alleviate the ambiguity. Text has been added to sections 3.3 and 3.4 to clarify that solely modelled data was used to calculate drivers (line 216), and that observations were used to extend these results to un-modelled data but that this extension was qualitative in nature (line 228). The word "modelled" has also been added to figure captions to make it clear from where the data displayed derives, and additional text stating that modelled water properties are based on transport weighed means

of the water parcels from a specified source water (lines 216, 352) have been added for clarity.

- Line 216: "In this study, all of the inputs into equation 2 are derived from the Lagrangian simulations."
- Line 228: "Observations are used in this study to extend the discussion of the drivers of biogeochemical variability beyond tracers available in LiveOcean. Tracers that exist in both the model (Section 3.1) and the collated observations are used as tools to qualitatively connect un-modelled tracers to the variability and attribution results in this study."
- Line 216, continued: "Values for Jiyear are the sum of water parcel volumes from a water source over a year and values for Piyear are the volume weighted mean properties from a source over the same period."
- Line 352: "To supplement this analysis, modelled variability in annual transport weighted mean source water properties (Fig. 7) was evaluated."

In addition, I find it unclear what observations were used to produce Fig. 5., i.e. if the averaging was performed on data close to the source lines or within regions, and if so how the spatial variability within those regions was considered. One can expect water properties to differ substantially from the westernmost point to the easternmost dark-blue point on the subpanel of Fig. 1. The observations chosen to produce Fig. 6 (previously Fig. 5) include all the points shown in the inset of Fig. 1, divided into water sources according to the definitions in Table 1. This has been clarified in the caption of Fig. 6.

With regards to the selection of observations to represent each of the water sources we agree that the offshore observations used cover too broad a region. A new cutoff of 1000 km from the shore has been applied in this draft, according to the definition of the California Current's spatial range in Hickey (1979), as opposed to the longitudinal cutoff used previously (described in line 243). While this new cutoff substantially reduces the number of observations in the two offshore water sources, their mean properties and standard deviations do not change significantly.

- Line 243: "To encompass the northern CCS, only observations shallower than 500 m, within 1000 km of the coast (Hickey, 1979), and between 40 and 50.8 °N were kept (Checkley and Barth, 2009)."
- 3.1 I have a concern regarding terminology. The authors refer to the different sources as "water masses" (e.g. L290). However, those sources are defined based on the properties at fixed lines, rather than by distinct water masses with coherent properties. The authors discuss the similarity between some of the source waters at L290-295. I recommend revising the terminology throughout the manuscript to "water sources" instead of "water masses". Replaced with "source water" or "water sources" throughout.
- 3.2 Related to this, I suggest that the authors comment on the similarity of source waters noted at L290-295 and discuss the meaning and implication of this similarity. This similarity

was explored further by checking if the similarity is persistent if the data is split into smaller latitudinal bands and if observations in the two sources vary similarly in time and space. The two water masses remain similar in time and latitudinal space, but differ more so in depth (line 331).

- Line 331: "These similarities persist in time and north-south location (based on 1° latitude binning), but some differences in the properties of the two water sources with depth are present. Between 100 m and 300 m the CUC and offshore deep water diverge more in many biologically influenced properties (DO, NO3, NO2, DIP, dissolved silicon (DSi), TA). This result may suggest that the CUC and offshore deep sources are made up of similar water mass mixtures, but that more respiration occurs in one source, the CUC in this case. Shallow source waters differ more-so: offshore surface water is more oxygen rich and has significantly higher aragonite and calcite saturation than shelf source waters. Among shelf waters, north shelf water is less spicy, while south shelf water exhibits a higher TA."
- 4. While the manuscript presents a separation into the upwelling vs downwelling season, the analysis stemming from this separation could be further developed. An attribution of the drivers per season, similar to what is presented in Fig. 7, would help discuss the impact of varying length of the upwelling and downwelling season (Fig. 2) on water properties. The precursor to this paper, Beutel & Allen (2024), goes into detail on the seasonal differences in water sources and their physical properties to the Salish Sea. Reference to the seasonal results of this paper are now stated more explicitly in the text (as detailed in response to general comment 1.2). What Beutel & Allen (2024) does not include is an analysis of the drivers of variability. A figure showing the drivers of interannual biogeochemical variability on seasonal scales has been included in the supplement (Fig. S12) and the implications of these seasonal differences are discussed (lines 396 and 419).
  - Line 396: "On seasonal scales, the relative importance of water sources to variability shift closer to their volume contributions (Fig. 3; Beutel and Allen (2024)). South shelf and south brackish water are very important to variability during downwelling, and are much smaller sources of variability during upwelling, when the contribution of offshore deep and north shelf water increase substantially (Fig. S12). The CUC is an important contributor to variability year-round."
  - Line 419: "However, Beutel and Allen (2024) also revealed a seasonal difference in the connection of JdF inflow to the inner basins, with parcels being much more likely to reach the inner basins during periods of upwelling. Thus, it follows that source waters more important to variability during upwelling: offshore deep, north shelf, and CUC water (Fig. S12), may be disproportionately influential on the biogeochemistry of the interior of the Salish Sea."

**Specific comments**

This is really only a suggestion, but I found the title not very engaging, and would suggest finding a more engaging title. Maybe putting forward the variability in water properties, with something like "Water property variability in a semi-enclosed sea dominated by dynamics,

modulated by properties". Title changed to "Water property variability into a semi-enclosed sea dominated by dynamics, modulated by properties".

Throughout the manuscript, best practice would be to provide the most recent citation first when referring to multiple papers. Fixed throughout.

L64-66: I suggest supporting this statement with a reference or with a figure. Beutel & Allen (2024) added to support the statement on the relationship between upwelling and downwelling to interannual variability (line 62), Checkley & Barth (2009) added to support the statement on the extent of the CCS (line 65).

- Line 62: "The contrasting properties of downwelled (less dense, fresher, more oxygen-rich, and nutrient- and DIC-poor) and upwelled water account for much of the seasonal variability in JdF inflow (Masson, 2006). However, these differences alone do not explain interannual variability (Beutel and Allen, 2024)."
- Line 65: "The Salish Sea is located at the northernmost end of the California Current System (CCS), an eastern boundary current system located between the North Pacific Gyre and the western coast of North America, spanning ~50 °N (Northern Vancouver Island, Canada) to ~15-25 °N (Baja California, Mexico) (Checkley and Barth, 2009)."

L81: For uniformity, I suggest including the range of values for oxygen as well. Oxygen ranges for the PEW (19-47  $\mu$ mol/kg) and PSUW (204–279  $\mu$ mol/kg) based on the OMP endmembers in Bograd et al. (2019) have been added to the text (lines 80 and 91).

- Line 80: "The CC predominantly carries Pacific Subarctic Upper Water (PSUW), a relatively cold (3°-15° C), fresh (32.6–33.7 psu), and oxygen-rich (204–279 μmol kg-1) water mass originating from the surface waters of the North Pacific (Bograd et al., 2019; Thomson and Krassovski, 2010)."
- Line 91: "The CUC carries PEW, a relatively warm (7°-23° C), saline (34.5–36.0 psu), oxygen poor (19–47 μmol kg-1), and nutrient-rich water mass originating from mixing in the equatorial Pacific (Bograd et al., 2019; Thomson and Krassovski, 2010)."

L118: I believe we should read "Resolution gradually decreases" and not "increases". Fixed.

L169: Providing an histogram of the time it takes for particles to cross the different boundaries would help convince the reader that 100 days is sufficient, in addition to providing useful information about the dynamics of the region. Such a figure could go in the supplementary material. It is now clarified in the text (line 176) that the particle crossing time of 100 days was chosen based on the findings in Beutel & Allen (2024), this is reinforced by the timing histogram added to the supplement of this paper (Fig. S8) which shows that almost all parcels have reached the completely their trajectory within 100 days. The small number of lost parcels (~1%, line 189) demonstrates that 100 days us sufficient for the grand majority of parcels to complete their trajectory.

• Line 176: "Particles were continuously released over the analysis period, with each run including an additional 100 days without particle seeding to allow particles

sufficient time to travel between boundaries based on histograms of crossing times in Beutel and Allen (2024) and checked again in this study (Fig. S8)."

In addition, how does the advection time compares with the duration of upwelling and downwelling events, and should a delay between the source water property definition and measurement at JdF be considered. While the typical parcel advection time (8-60 days) is small compared to the length of the average upwelling and downwelling period it is likely that many parcels reaching JdF at the beginning of these periods are impacted by the preceding event. This consideration has been added to the text (line 301). As noted above, a histogram of parcel advection times has been added to the supplement. As the focus of this paper is on interannual variability, and the property definitions provided in Table 2 are multi-year averages, it is not believed that a delay is required.

• Line 301: "As is the case with any continuous process, it should be noted that the dynamics of a given year will be impacted by the one preceding it. Typical parcel advection time differs between source waters (Fig. S8), ranging between 8 days for south brackish water and 60 days for offshore deep water. As such, some slower advecting source waters may be brought into the vicinity of JdF during a preceding period. For example, it is possible that parcels originating from the CUC at the beginning of downwelling or the fall transition were actually brought onto the shelf during upwelling (note the slightly higher contribution of CUC water in November and December than in January and February, Fig. 3e)."

L171: Given the chaotic nature of particle trajectories, which will look different for different experiments due to numerical error, how will using three separate runs per analysis affect the results? For example, one specific parcel might not provide exactly the same oxygen concentration at its source location if the experiment was run multiple times, and could in this case lead to associate oxygen and nitrate concentrations from different sources. Running multiple experiments for the same tracer would allow to diagnose the uncertainty associated with this method. I suggest presenting the results from such tests. The exclusion of subgrid-scale mixing in Ariane removes this chaotic nature, such that experiments are completely repeatable. This has been clarified in the text (line 159).

• Line 159: "The exclusion of subgrid-scale mixing enables repeatable runs and backwards tracking in Ariane simulations, meaning that multiple experiments over the same domain and time will have the same results (Van Sebille et al., 2018; Blanke and Raynaud, 1997)."

L176: Please provide the proportion of tidally-pumped parcels. The percentage of total transport that is considered tidally pumped (72%) has been added to the text (line 187).

• Line 187: "Water that is already within the analysis domain (lost parcels, representing ~1% of seeded parcels) or tidally pumped water (representing ~72% of seeded parcels) are excluded from these estimates."

In addition, are there parcels moving inland from the initialization line, and what is their

proportion? These would not be the same as lost particles between the initialization line and sources lines, but could rather come from recirculation or surface currents? I understand that the current method, whereby parcels are saved when they reach a line, might not allow to answer this question, but tests with a smaller number of particles could. It has been clarified in the text (line 168) that parcels are only seeded where the direction of transport is towards the analysis domain. Recirculating particles are captured in "loop" flow, while parcels that are in theory pushed back and forth over the initialisation section due to tides (in practice these are seeded, return to the initialisation section within one or two tidal cycles, and are reseeded) are captured in "tidally-pumped" flow and removed.

• Line 168: "Parcels are distributed across the initialization section at each time-step proportional to the transport (where transport through a cell q is equal to the velocity through a cell multiplied by its area,  $q=u\times A$ ) in each model grid cell where the direction of transport is towards the analysis domain (westward in Fig. 1)."

L191-195: Since this can be seen in the figures and tables and does not affect the method, I suggest removing. Removed.

L210-212: When reading the first time, it is not very clear how this classification will be used. I suggest more explicitly stating that these are the definitions used for the source water masses. That the classification will be used to divide observations into water sources has now been explicitly stated in the text (line 259). Moreover, please specify the maximum depth along the north boundary, to confirm that all waters sourced there can be defined as shallow waters. While the northern boundary does extend to the ~2000 m isobath 85% of water originating from the northern slope originates from depths shallower than 200 m. As such, while the northern boundary is not purely shallow, the parcels originating from it can be described as so. This intricacy has been clarified in the text (line 195).

- Line 259: "Further division into source waters (north shelf, offshore surface, south brackish, south shelf, CUC, offshore deep, and domain; Table 1) was based on trajectories identified in this study (Sec. 4.1) and Beutel and Allen (2024), and property-property diagrams of temperature, SA, NO3, and TA."
- Line 195: "Despite the extension of the northern boundary to near the 2000 m isobath, water parcels originating along that boundary are defined as shallow as they predominantly (85% of transport) intersect the boundary at depths shallower than 200 m."

Figure S3: Panels a-o do not highlight a clear separation. I suggest focusing on the S-property plots. The box plots are intended to highlight that very little change in the mean and interquartile range of the source water properties occurs due to the  $\pm 0.2$  g/kg change in salinity boundary, despite the S-property plots suggesting that a larger change may occur due to the proximity of the CUC versus south shelf division to the high density core of parcels. This has been emphasized in the text (line S68) and added to the caption of figure S3.

• Line S68: "Most significant is the increased range of DO and NO3 in the CUC and south shelf waters in response to an increase in division to 33.9 g kg-1, which cuts into

- the high salinity core of the CUC (Fig. S3r and q); however, the mean and range in NO3 and DO in these water sources remains relatively unchanged (Fig. 3a-j)."
- Relevant Fig. S3 caption line: "The box plots highlight that property ranges in each source water change little according to a  $\pm 0.2$  g kg-1 change in salinity cutoff."

**Fig. 1:**

a. In the subpanel, I suggest using a paler color for the 2000 m, since it can be confused with the coast for people not used to looking at this region. The colouring for the bathymetric contours and the coast have now been switched, with the coast in black and the bathymetry in grey.

b. I believe the initialization line is red, not brown. The colour of the initialization line has been changed to green to increase contrast.

L221: "grey bars in Fig. 2b, Bakun..." The sentence has been reworded as suggested (line 257).

- Line 257: "A combination of upwelling estimates were used to identify the length of these periods: meridional velocity measurements at moorings A1 and CE07 (Fig. 1; maintained by the DFO and the Ocean Observatories Initiative (OOI), and available from 2013-2020 and 2015-present, respectively), spring and fall transition timing (upwelling highlighted by yellow bars in Fig. 2b; Hourston and Thomson (2024), available from 1980-present), and the Bakun Index at 48 °N (upwelling highlighted by grey bars in Fig. 2b; Bograd et al. (2019); Bakun (1973), available from 1967-present)."
- Eq. 2: I would like to invite the authors to clarify Eq. 2. If we think of fields P and J as being separable into a mean and a yearly anomaly term (P = Pbase + Pyear), we would have four terms, including a crossed PbaseJbase term. As Pbase and Jbase are constant terms they are cancelled out when converting from Eq.1 to Eq.2. A step-by-step of this calculation has been added to the supplement (Section S4).

L254: Fig. 3a does not show the looped parcels. The authors should point to Fig. S6 or add the looped particles to Fig. 3. The text now points to Fig. 3b (Fig. 4b in the previous draft) as a reference for the inflow contribution of loop water and its seasonal variation as opposed to Fig. 3a.

L260: What is the full density range of the water at JdF? Having an idea of the vertical structure of the section would help visualizing what dynamics we are looking at. A Hovmöller diagram of the climatological mean monthly density with depth at the initialisation boundary in JdF has been added to the text (Fig. 4).

Fig. 3: I suggest adding the percentages on Fig. 3a if possible, to relate the result more easily with those of Fig. 7. Percentage contributions have been added to the bars of Fig. 3 – the caption has been updated accordingly.

- Fig. 4: I suggest showing the total volume flux. Visualizing the seasonal variability of the total volume flux would help seeing how the different source waters contribute to the total variability. Figure 4a (previously another visualisation of the variability in number of upwelling and downwelling days) has been changed to a panel showing total volume flux.
- Fig. 5: This is really just a suggestion, but, since the objective of this figure is to compare the water properties of different water sources, I suggest inverting the axis of the figure (hence show different water sources in x and variables in y) to ease the comparison. Unfortunately, given the large number of tracers being compared, Fig. 5 becomes too busy or too long to fit on one page when the axis is inverted.
- L335: For 2018, it appears that property variability plays a larger role for all properties, not only TA-DIC. Maybe this is worth discussing. In the ten years studied, the volume transport in 2018 is not extreme in any way it has an "average" total transport and percentage transport from each water mass. While the mean NO3, [TA-DIC], and DO values in 2018 are also not the most extreme highs or lows in the ten years studied, they are relatively close to said extremes. The combination of these two factors makes 2018 stand out in terms of property driven variability in NO3, [TA-DIC], and DO. The fact that 2018 stands out, despite not being a particularly extreme year, is in itself interesting. A short discussion on the occasional disconnect between years with extreme properties and those with larger property driven variability has been added to improve the clarity of how to interpret the attribution results (line 379).
  - Line 379: "High property driven variability in a given year does not directly align with interannual extremes in said properties. For example, in 2017 the [TA-DIC] concentration in south shelf water is more extreme than it was in 2016; however, property variability from south shelf water is a smaller driver in 2017 because higher than typical transport from shallow water sources overshadow the property impact."
- Fig. 6: I suggest adding the standard deviation on the seasonal means on the right panels, to provide information on how the intra-seasonal variability compared with the separation between seasons. The markers representing upwelling and downwelling means in the right panels have been changed to orange (upwelling) and blue (downwelling) box and whisker plots to show the range of values experienced in those periods in each of the water sources.
- L440: Near the surface, N\* can be affected not only by denitrification, but also by nitrogen deposition (e.g. https://doi.org/10.5194/bg-5-1199-2008) and mixing. I suggest that the authors consider whether and how this could affect their interpretation of the N\* results. Text addressing N\* as a measure of denitrification has been added (line 508).
  - Line 508: "It should be noted that deviations from the Redfield ratio (and thus N\*) can occur due to processes outside of denitrification, such as atmospheric deposition of nitrogen rich material, differing rates of nitrogen and phosphorus uptake or remineralisation, and nitrogen fixation (Landolfi et al., 2008). It is possible that the negative N\* found in the shallow water masses is due to the preferential

remineralisation of total organic phosphorus or the preferential uptake of NO3 in the surface layer, as opposed to strong denitrification at the shelf bottom."

L500-504: I suggest to move this to method section, to clarify how the data was used and how the water source properties were defined. Moved to the methods (line 186) and removed from limitations.

L507: I suggest to mention this earlier, in section 3.2. The potential slowing of parcels due to the lack of subgrid-scale mixing has been added to the methods (line 156), but has been kept in the limitations as well.

• Line 156: "Large turbulent eddies are resolved SalishSeaCast, a 500~m resolution model of the Salish Sea that uses LiveOcean output in its JdF boundary, such that the lack of explicit subgrid-scale mixing does not significantly alter or slow transport, even in mixing hot-spots (Allen et al., 2025; Stevens et al., 2021). Given LiveOcean's high resolution in the model domain, particularly in higher mixing areas near the entrance to JdF, similarly low impact is expected for the simulations in this study."

**Reviewer 2**

Beutel et al. present a practical approach to disentangle the effects of variable volume imports from effects of variable import properties on the variability of biogeochemical properties in the Salish Sea – a marginal sea of the Pacific Ocean. By backtracing water parcels in a coupled ocean-circulation biogeochemical model biogeochemical variability in the Sea could be traced back to imported contributions of distinct Pacific water masses.

The manuscript is well structured and clearly written. It is of interest to a wide audience (beyond those interested in the Salish Sea) also because it showcases how causes of variablity in estuary systems may be explored by combining observations with coupled ocean-circulation biogeochemical models (I especially liked Figure 7). Since this is a relevant question in many other coastal regions potentially affected by climate change the paper has both high scientific and societal relevance. I suggest, however, minor revision.

My suggestion is to clarify / elaborate on how the observations support conclusions. I think this could strengthen the manuscrip because the combination of simulated trajectories with observed concentrations may raise some questions. For example:

Ln. 367 in the discussion "... Combining observed water mass properties with model results can help reveal drivers of variability in biogeochemical tracers not explicitly represented in the model ...." This line has been changed to "Matching observations to source waters can help reveal drivers of variability in biogeochemical tracers not explicitly represented in the model" (line 429).

and

Ln. 531 in the conclusion "... Water mass contributions to JdF inflow, and the modelled and observed biogeochemical properties of these water masses, highlight the diverse drivers of

interannual variability in tracer flux. ..." This sentence has been kept the same, it is the hope that the explanation and detail added to the methods section (outlined below) will help clarify how the observations were used.

raise the questions: why are observed water mass properties needed? Is it because the model's water mass properties are inaccurate? If so, does this imply that the model physics are biased? And if that is the case, can we trust the backtracking? Observations were used in this study to extend the discussion of the drivers of biogeochemical variability beyond tracers available in the model, not to replace those in the model output. Model evaluations separated by source, as well as those previously reported in Xiong et al. (2024), suggest that salinity, temperature, DO, NO3, TA, and DIC fields are sufficiently close to measurements to support their use in this study (lines 134-140). The attribution of drivers was only done using model output, the aim of looking at observations of the modelled and un-modelled tracers is to identify where some un-modelled tracers may behave like modelled ones. For example, in the case of nutrients, NO3 and DIP varied similarly between water masses whereas DSi did not. This comparison of observations allowed for a discussion of the factors that are likely important to DIP and DSi despite neither of them being modelled. This use of observations is now included in the methods, both in Section 3.3 (line 214) where it is explained that observations were not used in the attribution calculation, and in Section 3.2 (line 228) where the use of observations is outlined.

- Line 214: "In this study, all of the inputs into equation 2 are derived from the Lagrangian simulations. Values for Jiyear are the sum of water parcel volumes from a water source over a year and values for Piyear are the volume weighted mean properties from a source over that same period. Where observations are available within each water source with sufficient spatial and temporal coverage to assess variation with confidence, it is possible to combine observed mean Pi with simulated Ji. This application of equation 2 was not examined in this paper, but is an interesting potential application for future study in areas such as the Newport Hydrographic Line (NHL) where observations are collected biweekly (Risien et al., 2024)."
- Line 228: "Observations are used in this study to extend the discussion of the drivers of biogeochemical variability beyond tracers available in LiveOcean. Tracers that exist in both the model (Section 3.1) and the collated observations are used as tools to qualitatively connect un-modelled tracers to the variability and attribution results in this study."

Further, Lagrangian Backtracking is a very common technique. It has been used extensively in the past to analyze water mass composition / origins of water masses. Maybe cite some exemplary (most pioneering or most sensational ...) papers of this field. The following references to papers that use Lagrangian backtracking to quantify water sources have been added to the description of Lagrangian tracking in the methods (line 151). This exemplary reference includes the following:

- Brasseale, E. and MacCready, P (2025), Seasonal Wind Stress Direction Influences Source and Properties of Inflow to the Salish Sea and Columbia River Estuary, *Journal of Geophysical Research: Oceans*, **130**, doi: 10.1029/2024JC022024.
- Chouksey, M., A. Griesel, C. Eden, and R. Steinfeldt (2022), Transit Time Distributions and Ventilation Pathways Using CFCs and Lagrangian Backtracking in the South Atlantic of an Eddying Ocean Model. *J. Phys. Oceanogr.*, **52**, 1531–1548, doi:10.1175/JPO-D-21-0070.1.
- deBoisséson, E., V. Thierry, H. Mercier, G. Caniaux, and D.
  Desbruyères (2012), Origin, formation and variability of the Subpolar Mode Water located over the Reykjanes Ridge, *J. Geophys. Res.*, 117, C12005, doi:10.1029/2011JC007519.

Ln. 8: "... Observations in the region provide insight into water mass processes not resolved by the model, including denitrification and ..." is confusing because, at this stage, it is unclear what model is referred to. This has been reworded to "Observations in the region provide insight into source water processes not resolvable in the Lagrangian simulations, including denitrification and trace metal supply."